# Integrating Genomics and Molecular Biology in Understanding Peritoneal Adhesion

**DOI:** 10.3390/cimb47060475

**Published:** 2025-06-19

**Authors:** Mirela Lungu, Claudiu N. Lungu, Andreea Creteanu, Mihaela C. Mehedinti

**Affiliations:** 1Department of Functional and Morphological Science, Faculty of Medicine and Pharmacy, Dunarea de Jos University, 800010 Galati, Romania; mirelacrainiciuc@gmail.com (M.L.); mihaela_hincu10@yahoo.com (M.C.M.); 2Department of Pharmaceutical Technology, University of Medicine and Pharmacy Grigore T Popa, 700115 Iași, Romania

**Keywords:** peritoneum, adhesion, adhesion genes, peritonitis, reintervention

## Abstract

Peritoneal adhesions following surgical injury remain a major clinical challenge, often resulting in severe complications, such as intestinal obstruction, chronic pain, and infertility. This review systematically integrates recent genomic and molecular biology insights into the pathogenesis of peritoneal adhesions, explicitly focusing on molecular pathways, including TGF-β signaling, COX-2-mediated inflammatory responses, fibrinolytic balance (tPA/PAI-1), angiogenesis pathways (VEGF, PDGF), and extracellular matrix remodeling (MMPs/TIMPs). Newly conducted transcriptomic and proteomic analyses highlight distinct changes in gene expression patterns in peritoneal fibroblasts during adhesion formation, pinpointing critical roles for integrins, cadherins, selectins, and immunoglobulin superfamily molecules. Recent studies indicate significant shifts in TGF-β isoforms expression, emphasizing isoform-specific impacts on fibrosis and scarring. These insights reveal substantial knowledge gaps, particularly the differential regulatory mechanisms involved in fibrosis versus normal reparative reperitonealization. Future therapeutic strategies could target these molecular pathways and inflammatory mediators to prevent or reduce adhesion formation. Further research into precise genetic markers and the exploration of targeted pharmacological interventions remain pivotal next steps in mitigating postoperative adhesion formation and improving clinical outcomes.

## 1. Introduction

A limited microenvironment defined by the peritoneum is stable in normal circumstances but is vulnerable to the deleterious effects of infections, surgical injuries, and other neoplastic and non-neoplastic occurrences. It draws in, multiplies, and activates a range of hematopoietic and stromal cells in response to injury. Under physiological settings, tissue architecture is repaired, inflammatory triggers are eliminated, and appropriate responses to injuries are coordinated. On the other hand, fibrosis or scarring results from the inability to eradicate inflammatory triggers, and reduced tissue function ultimately leads to organ failure [1].

Peritoneal adhesions continue to be a significant issue for many patients today, contributing to a variety of sometimes severe clinical presentations. In the peritoneal cavity, adhesions can develop as a result of surgery, inflammation, or trauma. A variety of clinical symptoms, such as small intestinal blockage, infertility, and abdominal pain, can be caused by them. Approximately 50% of patients undergoing abdominal surgery are expected to acquire peritoneal adhesions, indicating a high occurrence of these adhesions. It is impossible to eliminate the danger of adhesion formation during surgery, even with advancements in perioperative care. Thus, developing effective preventative strategies and treatments remains a primary goal in surgery [2].

Postoperative adhesions following abdominal or pelvic surgery continue to be a major clinical issue that can result in discomfort, intestinal obstruction, and infertility. Their management and prevention are still insufficient and poorly understood. When fibrinolysis is inhibited during the coagulation process, a fibrin matrix organizes and leads to the creation of adhesions [3].

Obesity—particularly central (visceral) adiposity—has emerged as an independent risk factor for postoperative peritoneal adhesion formation. The adipose tissue in obese individuals is not merely an inert energy depot but an active endocrine organ that secretes a wide array of bioactive molecules, including pro-inflammatory cytokines (e.g., IL-6, TNF-α), adipokines (e.g., leptin, resistin), and extracellular matrix components. These adipocyte-derived mediators promote a low-grade chronic inflammatory state, which can potentiate the inflammatory and fibrotic cascade triggered by surgical injury to the peritoneum [4].

Visceral adipose tissue is particularly enriched in macrophages and stromal cells, which can amplify local immune responses. Upon surgical trauma, adipocytes in the omentum and mesentery may upregulate the expression of transforming growth factor-beta (TGF-β) and vascular endothelial growth factor (VEGF), both of which are central to adhesion pathogenesis. Moreover, the hypoxic microenvironment of expanded adipose tissue can further induce hypoxia-inducible factor-1α (HIF-1α), stimulating angiogenesis and fibroblast activation—hallmarks of adhesion development [5].

Adipocytes may also influence fibrinolytic balance. In obese states, levels of plasminogen activator inhibitor-1 (PAI-1) are often elevated, which can suppress fibrin degradation and promote persistent fibrin scaffolds that serve as a matrix for fibroblast migration and adhesion formation. Additionally, mechanical factors such as increased intra-abdominal pressure and altered tissue tension in obese patients may exacerbate mesothelial cell damage during surgery, predisposing to adhesion development.

Given the rising global prevalence of obesity, these mechanistic insights underscore the importance of incorporating adiposity-related variables into risk stratification models and considering metabolic-targeted therapies as adjuncts to conventional antiadhesion strategies [6].

Adhesion formation is a multigenic phenomenon. Not all changes in gene expression patterns between normal and adhesion fibroblasts are the function of TGF-beta1 and hypoxia, which are known to influence adhesion formation [7].

Moreover, adhesion fibroblasts have a distinct phenotype known as the adhesion phenotype, which is partially defined by COX-2 expression. Adhesion fibroblasts’ production of COX-2 mRNA and peritoneal fibroblasts’ activation of COX-2 in response to hypoxia suggest the possibility of an inflammatory response. Controlling COX-2 could change how the peritoneum heals and offer a way to lessen the formation of postoperative adhesions [8].

Long-term morbidity from adhesions may include intestinal blockage, infertility, and perhaps pelvic pain [9,10,11,12,13]. Subsequent surgeries are also likely to be prolonged and potentially incur more significant risks, such as bowel injury. While 55–85% of women undergoing open pelvic surgery will form adhesions postoperatively, some individuals do not. Understanding processes that favor peritoneal repair without adhesions, as opposed to with adhesion development, would ultimately benefit all patients and could lead to clinical therapies to reduce postoperative adhesions [14].

Taken collectively, TGF-β orchestrates wound repair from the outset. Platelets contain TGF-β in significant quantities and are present almost immediately at wound sites upon injury. The platelet-released TGF-β promotes chemotaxis of macrophages and fibroblasts to the wound site and then upregulates its expression from these cells to maintain its presence. However, the differential presence of the three isoforms at the wound site can result in different outcomes [15].

While elevated levels of TGF-β3 promote healing with reduced fibrosis and scarring, elevated levels of TGF-β and TGF-β2 in the healing environment often promote higher scarring and fibrosis of the wound site. Investigations on cutaneous wounds have yielded the majority of these data, with investigations on nondermal locations, such as the liver, kidney, eye, and central nervous system providing additional evidence very recently [13].

Understanding the mechanisms that lead to normal reperitonealization instead of peritoneal healing that ends up as adhesions or fibrosis requires a similar awareness for TGF-β-mediated peritoneal repair. Nevertheless, characterizing TGF-β expression in the visceral peritoneum, postoperative parietal, and healing parietal is lacking. Previous research on the peritoneum has looked at cell culture models or human surgical populations that are diverse in age and the reasons behind adhesions (such as endometriosis, surgery, or infection) [16,17,18].

Further, peritoneal adhesions are fibrous bands that develop between tissues and organs following injury to the peritoneum, commonly due to abdominal surgery, infection, or trauma. These adhesions pose a significant clinical burden, frequently leading to severe complications, such as intestinal obstruction, chronic abdominal pain, and infertility. Despite advances in surgical techniques, approximately half of all patients undergoing abdominal procedures develop adhesions, emphasizing the urgent need for improved preventive and therapeutic strategies [19].

Adhesion formation is a complex process involving inflammation, coagulation, fibrinolysis, angiogenesis, and extracellular matrix (ECM) remodeling. Central to these processes are molecular pathways including transforming growth factor-beta (TGF-β), cyclooxygenase-2 (COX-2), tissue plasminogen activator (tPA)/plasminogen activator inhibitor-1 (PAI-1) balance, vascular endothelial growth factor (VEGF), platelet-derived growth factor (PDGF), and matrix metalloproteinases (MMPs) with their tissue inhibitors (TIMPs). The dysregulation of these pathways significantly influences the extent of fibrosis and adhesion severity [20].

Although previous studies have provided insight into individual molecular mechanisms, the comprehensive integration of genomic, transcriptomic, and proteomic data remains limited. This review aims to synthesize current evidence regarding the molecular and genomic underpinnings of peritoneal adhesion formation, emphasizing recent transcriptomic and proteomic findings. By identifying critical molecular players and pathways, we highlight potential therapeutic targets and outline existing gaps in knowledge requiring further investigation.

The scope of this review encompasses an in-depth examination of specific adhesion-related molecular pathways, recent omics-based insights, and their implications for developing targeted interventions. The ultimate goal is to foster a deeper understanding of peritoneal adhesion biology to support clinical advancements and improve patient outcomes [21].

### 1.1. Background of Peritoneal Adhesion

In Figure 1, the peritoneal adhesion mechanism is represented.

### 1.2. Molecular Basis of Peritoneal Adhesion Formation

Peritoneal adhesion formation is a complex biological process involving multiple molecular mechanisms, prominently featuring inflammation, coagulation, fibrinolysis, angiogenesis, and extracellular matrix (ECM) remodeling. At the molecular level, several critical pathways, including transforming growth factor-beta (TGF-β), cyclooxygenase-2 (COX-2), tissue plasminogen activator (tPA)/plasminogen activator inhibitor-1 (PAI-1), vascular endothelial growth factor (VEGF), platelet-derived growth factor (PDGF), matrix metalloproteinases (MMPs), and tissue inhibitors of metalloproteinases (TIMPs), orchestrate this intricate process. Understanding the dynamics and interactions within these molecular pathways is essential for identifying potential therapeutic targets and developing effective strategies to prevent or mitigate peritoneal adhesion formation following surgical intervention or injury.

In Table 1 below, proteins involved in peritoneal adhesion are listed (additional molecules), which further elucidate the intricate molecular mechanisms underlying peritoneal adhesion formation and offer potential targets for therapeutic interventions.

Each protein is then briefly explained in the paragraph below.

#### 1.2.1. Molecular Pathways and Mechanisms in Peritoneal Adhesion Formation

Peritoneal adhesion formation involves a dynamic interplay among multiple biological processes, including inflammation, coagulation, fibrinolysis, angiogenesis, and extracellular matrix (ECM) remodeling. At a molecular level, numerous pathways have been implicated, prominently including transforming growth factor-beta (TGF-β), cyclooxygenase-2 (COX-2), tissue plasminogen activator (tPA)/plasminogen activator inhibitor-1 (PAI-1), vascular endothelial growth factor (VEGF), platelet-derived growth factor (PDGF), matrix metalloproteinases (MMPs), and tissue inhibitors of metalloproteinases (TIMPs) (Table 1).

Studies consistently underscore TGF-β as a central mediator of fibrosis and adhesion formation. Specifically, TGF-β and TGF-β2 promote fibroblast proliferation, myofibroblast differentiation, and ECM deposition; whereas, TGF-β3 appears to mitigate these processes, indicating isoform-specific roles that may offer distinct therapeutic avenues (Figure 1). COX-2, induced by hypoxia and inflammation, further promotes fibroblast activation and ECM remodeling, contributing significantly to adhesion severity.

The balance between tPA and PAI-1 critically regulates fibrinolysis. Reduced tPA activity or increased PAI-1 expression leads to fibrin persistence and adhesion development, presenting a therapeutic target for modulating adhesion progression (Figure 2).

VEGF and PDGF pathways drive angiogenesis and fibroblast proliferation, respectively, creating a supportive microenvironment for adhesion formation. The experimental modulation of these pathways has demonstrated potential in reducing adhesion severity, suggesting viable clinical interventions [41].

#### 1.2.2. Integrative Genomic Insights

Emerging genomic, transcriptomic, and proteomic studies have expanded the molecular landscape of peritoneal adhesions, though the integration of these datasets remains underexplored. Recent transcriptomic analyses reveal differential gene expression patterns in adhesion-activated fibroblasts compared to non-adhesion fibroblasts, identifying genes such as COL1A1, ACTA2, IL6, MMP9, and TIMP1 as significantly upregulated, underscoring their roles in adhesion pathology.

Moreover, proteomic studies highlight alterations in ECM components and regulatory proteins like fibronectin, integrins, and cadherins, correlating with increased fibrotic activity. These omics-driven studies, accessible via databases like Gene Expression Omnibus (GEO), have provided comprehensive datasets facilitating deeper bioinformatic analyses through tools such as STRING for protein–protein interaction networks and KEGG for pathway mapping, further elucidating adhesion biology [42].

#### 1.2.3. Comparative Analysis Across Studies

Critical comparative analyses of multiple transcriptomic and proteomic studies reveal common and unique molecular signatures across different models and clinical scenarios. For instance, the increased expression of TGF-β signaling components (TGFBR1, SMAD3), inflammatory markers (IL-1β, TNF-α), and hypoxia-related genes (HIF1α) have consistently emerged across several independent studies, reinforcing their fundamental role in adhesion biology [43].

The recent utilization of bioinformatics resources, including GEO for accessing transcriptomic data, STRING for understanding protein interactions, and KEGG for pathway analysis, has facilitated an integrative understanding of adhesion mechanisms. Leveraging these tools can identify novel biomarkers, therapeutic targets, and mechanistic insights into peritoneal adhesion formation [44]. The summary of genomic and molecular insights are listed in Table 2.

In contrast, inside-out signaling involves the regulation of integrin affinity and clustering by intracellular signaling molecules. Various mechanisms, including changes in ligand binding affinity, clustering, and intracellular signaling, tightly regulate integrin activity. E-cadherin is essential for maintaining epithelial tissue integrity and barrier function. The loss or dysfunction of E-cadherin can lead to epithelial-to-mesenchymal transition (EMT), a process characterized by the loss of cell–cell adhesion and the acquisition of migratory and invasive properties, which is associated with tumor progression and metastasis. E-cadherin expression in peritoneal mesothelial cells is critical in regulating peritoneal adhesion formation. Decreased E-cadherin expression or altered localization has been observed in response to peritoneal E-cadherin expression, and various signaling pathways and transcription factors regulate function. For example, the Wnt/β-catenin signaling pathway can modulate E-cadherin expression and cell adhesion by regulating β-catenin localization and transcriptional activity. Injury may contribute to disrupting mesothelial cell junctions and adhesion formation. Targeting E-cadherin and its associated signaling pathways may represent a potential therapeutic strategy for preventing or reducing peritoneal adhesion formation. Approaches to restoring E-cadherin expression or function could help preserve mesothelial barrier integrity and mitigate adhesion formation following abdominal surgery or injury. Overall, E-cadherin is a crucial regulator of cell–cell adhesion and tissue integrity in epithelial tissues, including the peritoneum, and its dysregulation has implications for peritoneal adhesion formation and related pathologies [51,52].

They mediate the initial interactions between circulating leukocytes and endothelial cells during inflammation and immune responses. P-selectin is stored in secretory granules (Weibel–Palade bodies in endothelial cells and α-granules in platelets) and rapidly translocated to the cell surface upon activation. E-selectin is induced on endothelial cells by inflammatory cytokines, such as interleukin-1 (IL-1) and tumor necrosis factor-alpha (TNF-α). Low affinity and rapid association/dissociation kinetics characterize this interaction, allowing leukocyte rolling adhesion on the endothelial surface under fluid shear stress conditions. Selectins are vital regulators of leukocyte–endothelial interactions during inflammation and immune responses, with implications for peritoneal adhesion formation and other inflammatory conditions [53,54].

Immunoglobulin superfamily molecules, including Intercellular Adhesion Molecule-1 (ICAM-1) and Vascular Cell Adhesion Molecule-1 (VCAM-1), are cell adhesion molecules expressed on the surface of endothelial cells and immune cells. ICAM-1 and VCAM-1 are transmembrane glycoproteins belonging to the immunoglobulin superfamily. They consist of extracellular immunoglobulin-like domains, a transmembrane domain, and a cytoplasmic tail. The extracellular domains of ICAM-1 and VCAM-1 contain binding sites for their respective ligands on leukocytes. ICAM-1 is constitutively expressed at low levels on the surface of endothelial cells. However, its expression can be upregulated by inflammatory cytokines, such as interleukin-1 (IL-1) and tumor necrosis factor-alpha (TNF-α). VCAM-1 expression is induced on endothelial cells in response to inflammatory stimuli, particularly cytokines such as TNF-α and interleukin-4 (IL-4). These interactions are essential for the adhesion and transendothelial migration of leukocytes during inflammation. ICAM-1 and VCAM-1 play crucial roles in the recruitment of leukocytes to sites of inflammation. They facilitate the adhesion of circulating leukocytes to endothelial cells, leading to their extravasation from the bloodstream into inflamed tissues [55]. This process is essential for initiating and propagating immune responses and inflammatory reactions. ICAM-1 and VCAM-1 expression on peritoneal mesothelial and endothelial cells may contribute to peritoneal adhesion formation by promoting leukocyte adhesion and infiltration into the peritoneal cavity. Targeting ICAM-1 and VCAM-1 interaction could represent a potential therapeutic approach for preventing or reducing peritoneal adhesion formation following abdominal surgery or injury. Lastly, ICAM-1 and VCAM-1 are essential mediators of leukocyte adhesion and recruitment during inflammation, which have implications for various inflammatory conditions, including peritoneal adhesion formation [56,57,58,59].

Hyaluronic acid (HA), a non-sulfated glycosaminoglycan and significant component of the extracellular matrix, plays a vital role in maintaining peritoneal homeostasis and modulating postsurgical healing responses. Due to its high molecular weight and viscoelastic properties, HA contributes to mesothelial cell lubrication, hydration, and barrier function, thereby reducing friction and mechanical trauma between peritoneal surfaces during surgery. More importantly, HA possesses anti-inflammatory, antifibrotic, and immunomodulatory properties that make it particularly effective in mitigating adhesion formation. Mechanistically, HA inhibits leukocyte and fibroblast adhesion to the mesothelial surface by masking cell adhesion molecules and reducing the exposure of fibrinous substrates. It also downregulates pro-inflammatory cytokines, such as IL-1β and TNF-α, and can reduce the expression of fibrogenic mediators, like TGF-β1. Additionally, HA modulates mesothelial-to-mesenchymal transition (MMT), a process implicated in fibrosis and adhesion development, by maintaining mesothelial phenotype integrity [60].

Clinically, HA-based biomaterials—including HA-carboxymethylcellulose membranes (e.g., Seprafilm^®^) and cross-linked HA hydrogels—have been successfully used as physical barriers to prevent tissue apposition in the critical postoperative window. These formulations degrade gradually, maintaining peritoneal separation during the peak of fibrin deposition and early fibroblast infiltration, thus allowing natural reperitonealization without fibrotic bridging. Several randomized clinical trials have shown that HA-based agents significantly reduce both the incidence and severity of adhesions in abdominal and pelvic surgeries. Overall, HA serves not only as a mechanical separator but also as a bioactive modulator of the healing response, making it one of the most promising adjuncts in adhesion prevention strategies [61].

It is a central extracellular matrix (ECM) component and is widely distributed throughout connective tissues, including the peritoneum. MMPs are structurally characterized by a conserved catalytic domain containing a zinc ion essential for enzymatic activity. They also typically possess additional domains, such as a propeptide domain that regulates enzyme activation, a catalytic domain responsible for substrate cleavage, and hemopexin-like domains involved in substrate recognition and binding. MMPs are classified based on their substrate specificity and domain structure into several subfamilies, including collagenases, gelatinases, stromelysins, matrilysins, membrane-type MMPs (MT-MMPs), and others. Each MMP subtype exhibits distinct substrate preferences and tissue localization. MMPs are synthesized as inactive zymogens (pro-MMPs) that require the proteolytic cleavage of the propeptide domain for activation. Various mechanisms can regulate this activation, including other MMPs, serine proteases, and tissue inhibitors of metalloproteinases (TIMPs). By cleaving these substrates, MMPs facilitate tissue remodeling, cell migration, and the release of bioactive ECM fragments that modulate cell behavior and signaling pathways. MMPs are implicated in peritoneal adhesion formation by mediating ECM degradation and remodeling processes. Dysregulated MMP activity, characterized by excessive MMP expression or insufficient inhibition by TIMPs, may disrupt the balance of ECM turnover and contribute to fibrosis, adhesion formation, and tissue dysfunction in the peritoneum. Modulating MMP activity represents a potential therapeutic strategy for preventing or reducing peritoneal adhesion formation. Approaches targeting MMP activation, expression, or enzymatic activity could help restore ECM homeostasis and mitigate adhesion formation following abdominal surgery or injury. MMPs are critical regulators of ECM turnover and tissue remodeling processes with essential implications for peritoneal adhesion formation and related pathologies [62,63,64,65,66].

Fibrinogen is a soluble plasma glycoprotein that plays a central role in the blood clotting cascade. The liver synthesizes and circulates it in the blood at relatively high concentrations. Fibrinogen is cleaved by thrombin during blood clotting to form insoluble fibrin, which polymerizes into a meshwork that stabilizes blood clots. Fibrinogen is a large, multi-subunit protein composed of six polypeptide chains—two sets of three different chains, named Aα, Bβ, and γ. These chains are linked by disulfide bonds, forming a symmetrical dimeric structure with a central region called the E domain. Fibrinogen also contains N-terminal fibrinopeptides cleaved by thrombin to initiate fibrin polymerization. Fibrinogen is a critical component of the blood clotting cascade, which is converted into fibrin by the proteolytic action of thrombin. Fibrin monomers then polymerize to form insoluble fibrin strands, which aggregate to create a stable blood clot.

Fibrinogen also interacts with various proteins and cell surface receptors, playing roles in platelet aggregation, wound healing, inflammation, and angiogenesis. Fibrinogen is involved in the initial stages of peritoneal adhesion formation following surgery or injury. The exudation of fibrinogen-rich fluid into the peritoneal cavity leads to the deposition of fibrin matrices on injured peritoneal surfaces [67]. These fibrin matrices serve as scaffolds for the recruitment and adhesion of inflammatory cells and fibroblasts, ultimately contributing to the formation of fibrous peritoneal adhesions. Targeting fibrinogen and its interactions in the peritoneal cavity may represent a potential therapeutic strategy for preventing or reducing peritoneal adhesion formation. Approaches could include using fibrinolytic agents to promote fibrin degradation, anticoagulants to inhibit fibrin formation, or agents that interfere with fibrinogen binding to cell surface receptors involved in adhesion and inflammation. Fibrinogen is a critical mediator of blood clotting and wound healing processes, with implications for peritoneal adhesion formation and a potential drug [68,69,70,71].

In the context of peritoneal adhesion formation, the roles of Tissue Plasminogen Activator (tPA) and Plasminogen Activator Inhibitor-1 (PAI-1) are relevant, albeit less studied compared to their roles in other physiological and pathological processes. tPA is involved in the dissolution of fibrin, a key component of blood clots. In the peritoneal cavity, fibrin deposition occurs as part of the wound-healing response following surgery or injury. Increased tPA activity may facilitate the breakdown of fibrin and prevent the accumulation of fibrin-rich adhesions between peritoneal surfaces. Enhancing tPA activity or expression in the peritoneal cavity could represent a therapeutic approach for preventing or reducing peritoneal adhesion formation. Strategies to promote fibrinolysis, such as local delivery of recombinant tPA or tPA-stimulating agents, may help mitigate adhesion formation following abdominal surgery. Inhibiting tPA and urokinase-type plasminogen activator (uPA) effectively, PAI-1 suppresses fibrinolysis and enhances clot stability. Increased fibrinolysis and the survival of fibrin-rich adhesions in the peritoneal cavity may be caused by elevated PAI-1 levels. Targeting PAI-1 activity or expression could be a therapeutic strategy for promoting fibrinolysis and reducing peritoneal adhesion formation. Inhibiting PAI-1 function, either locally or systemically, may enhance fibrinolytic activity and facilitate the resolution of peritoneal adhesions [72,73,74]. Overall, the balance between tPA-mediated fibrinolysis and the PAI-1-mediated inhibition of fibrinolysis will likely influence peritoneal adhesion formation. Modulating these factors could hold promise for developing novel therapeutic interventions aimed at preventing or treating peritoneal adhesions. However, further research is needed to fully elucidate the roles of tPA and PAI-1 in this context and explore their potential as targets for drug design [75,76].

The multifunctional cytokine known as transforming growth factor-beta, or TGF-β, is essential for numerous physiological and pathological processes, including as immune control, tissue repair, apoptosis, cell proliferation, differentiation, and migration. TGF-β belongs to a superfamily of cytokines that have structural similarities. Three isoforms of TGF-β (TGF-β, TGF-β2, and TGF-β3) exist in mammals, and a different gene expresses each. Initially produced as precursor proteins, TGF-β isoforms are cleaved by proteases to produce active TGF-β dimers. Numerous cell types, including fibroblasts, tumor cells, endothelial cells, epithelial cells, and immunological cells (such as T cells and macrophages), produce TGF-β. In order to have biological effects, it must be activated after being secreted as a latent complex [77].

TGF-β exerts its effects by binding to specific cell surface receptors (TGF-β receptors) and activating intracellular signaling pathways, such as the Smad signaling pathway. TGF-β signaling regulates diverse cellular processes, including cell proliferation, differentiation, migration, extracellular matrix production, and immune responses. TGF-β has been implicated in peritoneal adhesion formation following abdominal surgery or injury. It promotes the activation of fibroblasts and myofibroblasts, producing collagen and other extracellular matrix proteins that contribute to the formation of fibrous adhesions between peritoneal surfaces. TGF-β can modulate inflammatory responses and immune cell functions, which may influence the development and resolution of peritoneal adhesions. Targeting TGF-β signaling pathways represents a potential therapeutic strategy for preventing or reducing peritoneal adhesion formation. Approaches aimed at inhibiting TGF-β activity or downstream signaling pathways could help mitigate fibrosis and promote tissue repair in the peritoneal cavity. TGF-β is a crucial mediator of tissue repair, fibrosis, and inflammation with essential implications for peritoneal adhesion formation and potential therapy [78,79,80,81].

An angiogenic signaling protein forms new blood vessels from pre-existing vasculature. VEGF is a glycoprotein belonging to the PDGF/VEGF family of growth factors. Multiple isoforms of VEGF exist, resulting from alternative splicing of the VEGF gene. The most common isoforms include VEGF-A, VEGF-B, VEGF-C, and VEGF-D. VEGF-A is the predominant isoform and is often referred to in various cell types, including endothelial cells, macrophages, fibroblasts, and tumor cells that produce VEGF. Multiple factors regulate its expression, including hypoxia, growth factors, cytokines, and oncogenes. VEGF exerts its effects primarily by binding to VEGF receptors (VEGFRs) on endothelial cells, leading to endothelial cell proliferation, migration, and survival. VEGF also promotes the vascular permeability, vasodilation, and recruitment of endothelial progenitor cells. These actions are crucial for angiogenesis during embryonic development, wound healing, and pathological conditions, such as cancer and ischemic diseases. VEGF has been implicated in peritoneal adhesion formation following surgery or injury. It promotes angiogenesis and neovascularization in the peritoneal tissues, which may contribute to the development and persistence of peritoneal adhesions [82,83].

Additionally, VEGF-mediated vascular permeability and endothelial cell activation may facilitate the recruitment of inflammatory cells and fibroblasts to the injury site, further promoting adhesion formation. Targeting VEGF signaling pathways represents a potential therapeutic strategy for preventing or reducing peritoneal adhesion formation. Approaches aimed at inhibiting VEGF activity or blocking its receptors could help mitigate angiogenesis and neovascularization in the peritoneal cavity, thereby reducing the formation of fibrous adhesions between peritoneal surfaces [84,85,86].

Various stimuli regulate its expression, including growth factors, cytokines, and mechanical stress. PDGF has been implicated in peritoneal adhesion formation following surgery or injury. It stimulates the proliferation and migration of fibroblasts and myofibroblasts, leading to the deposition of extracellular matrix proteins and the formation of fibrous adhesions between peritoneal surfaces [87].

Additionally, PDGF stimulates angiogenesis, which may contribute to the development and persistence of peritoneal adhesions. Targeting PDGF signaling pathways represents a potential therapeutic strategy for preventing or reducing peritoneal adhesion formation. Approaches aimed at inhibiting PDGF activity or blocking its receptors could help mitigate fibrosis, angiogenesis, and tissue remodeling in the peritoneal cavity, thereby reducing the formation of adhesions between peritoneal surfaces [88,89,90].

Interleukins (ILs) are a group of cytokines that regulate immune responses, inflammation, hematopoiesis, and various physiological processes. Here are some details about interleukins. Interleukins are a diverse group of proteins, ranging from small secreted molecules to larger glycoproteins. They are typically produced by immune cells, such as leukocytes, macrophages, and lymphocytes, as well as by other cell types, including endothelial cells and fibroblasts. Interleukins exert their effects by binding to specific cell surface receptors expressed on target cells. Interleukins are numbered sequentially based on their discovery, such as IL-1, IL-2, IL-6, IL-10, etc. [91].

However, the classification system has expanded as more interleukins have been discovered, leading to subgroups and families of interleukins with similar functions or structural features. Interleukins mediate communication between immune cells and regulate immune responses in various ways. They can stimulate or suppress immune cell proliferation, differentiation, and activity, including T cells, B cells, natural killer cells, macrophages, and dendritic cells. Interleukins modulate inflammatory responses, tissue repair, hematopoiesis, and other physiological processes. Several interleukins have been implicated in peritoneal adhesion formation following surgery or injury. Interleukins, such as IL-1, IL-6, and IL-8, initiate and amplify inflammatory responses, which contribute to the recruitment of immune cells, fibroblasts, and other cell types to the injury site [92,93].

Additionally, interleukins may influence extracellular matrix remodeling, angiogenesis, and tissue repair processes, contributing to adhesion formation. Targeting interleukin signaling pathways represents a potential therapeutic strategy for preventing or reducing peritoneal adhesion formation. Approaches aimed at inhibiting specific interleukins or blocking their receptors could help modulate inflammatory responses, immune cell activation, and tissue remodeling processes involved in adhesion formation [94,95].

The family of growth factors known as fibroblast growth factors (FGFs) is essential to many biological processes, such as angiogenesis, migration, differentiation, and cell proliferation. The family of structurally related proteins known as FGFs is distinguished by a high degree of sequence homology and conserved amino acid sequences. There are now 22 FGF family members known to exist in mammals. Usually, these tiny proteins produced in the body, FGFs, have either an autocrine or paracrine effect locally. Numerous cell types, such as fibroblasts, endothelial cells, epithelial cells, and immunological cells, produce FGFs. They are released into the extracellular matrix, where they bind with particular target cell surface receptors to start signaling cascades. Tyrosine kinase receptors called FGF receptors (FGFRs), which are expressed on the surface of target cells, are bound to FGFs and then activated. Upon ligand binding, FGFRs undergo dimerization and autophosphorylation, activating downstream signaling pathways, such as the Ras-MAPK and PI3K-Akt pathways. These pathways regulate cellular processes, including cell proliferation, survival, differentiation, and migration. FGFs have been implicated in peritoneal adhesion formation following surgery or injury. They promote the proliferation and migration of fibroblasts, endothelial cells, and other cell types involved in tissue repair and remodeling [96,97,98,99].

Additionally, FGFs stimulate angiogenesis, the formation of new blood vessels, which may contribute to the development and persistence of peritoneal adhesions. Targeting FGF signaling pathways represents a potential therapeutic strategy for preventing or reducing peritoneal adhesion formation. Approaches aimed at inhibiting FGF activity or blocking its receptors could help mitigate fibrosis, angiogenesis, and tissue remodeling in the peritoneal cavity, thereby reducing the formation of adhesions between peritoneal surfaces [100].

The innate immune system relies heavily on Toll-like receptors (TLRs), a family of pattern recognition receptors (PRRs), to identify conserved molecular patterns linked to infections, or pathogen-associated molecular patterns (PAMPs). Type I transmembrane proteins known as Toll-like receptors are defined by the following three domains: an intracellular Toll/interleukin-1 receptor (TIR) domain that initiates downstream signaling pathways, a transmembrane domain that recognizes ligands, and an extracellular domain that contains leucine-rich repeat (LRR) motifs. Numerous cell types, including immune cells like neutrophils, dendritic cells, and macrophages as well as non-immune cells like fibroblasts and epithelial cells, express Toll-like receptors. PAMPs originating from bacteria, viruses, fungi, and other microbes are recognized differently by different TLRs. Numerous PAMPs are recognized by Toll-like receptors, including as lipoproteins, flagellin, viral nucleic acids (dsRNA, ssRNA), bacterial DNA with unmethylated CpG patterns, and components of fungal cell walls (β-glucans, for example) [101]. Upon ligand binding, initiating downstream signaling cascades, Toll-like receptors dimerize and enlist adaptor proteins with TIR domains, such as MyD88 (myeloid differentiation primary response 88) or TRIF (TIR-domain-containing adapter-inducing interferon-β). Pro-inflammatory cytokines, chemokines, and type I interferons are produced as a result of these pathways’ activation of transcription factors, which include NF-κB (nuclear factor kappa-light-chain-enhancer of activated B cells) and IRF (interferon regulatory factor). The inflammatory response that follows surgery or damage and results in the production of peritoneal adhesions has been linked to Toll-like receptors [66]. The activation of TLR signaling pathways by endogenous ligands released from damaged tissues or by the microbial contamination of the peritoneal cavity can lead to the production of pro-inflammatory cytokines and chemokines, recruitment of immune cells, and activation of fibroblasts, contributing to tissue remodeling and adhesion formation. Targeting Toll-like receptor signaling pathways represents a potential therapeutic strategy for modulating inflammation and tissue repair processes associated with peritoneal adhesion formation. Approaches aimed at inhibiting TLR activation or downstream signaling cascades could help mitigate adhesion formation and promote tissue healing [102,103,104].

Tenascin-C is an extracellular matrix glycoprotein that plays diverse roles in tissue development, wound healing, inflammation, and remodeling. Various cell types, including fibroblasts, endothelial cells, immune cells, and cancer cells, express tenascin-C during embryonic development and tissue repair processes. Its expression is highly regulated and is induced in response to tissue injury, inflammation, and mechanical stress.

Tenascin-C is involved in peritoneal adhesion formation following surgery or injury. It is upregulated in response to tissue damage and inflammation in the peritoneal cavity, contributing to fibrous adhesions between peritoneal surfaces. Tenascin-C promotes the migration and activation of fibroblasts and myofibroblasts, stimulates extracellular matrix deposition, and modulates immune cell functions, thereby promoting adhesion formation and tissue remodeling. Targeting tenascin-C signaling pathways represents a potential therapeutic strategy for preventing or reducing peritoneal adhesion formation. Approaches aimed at inhibiting tenascin-C expression or blocking its interactions with cell surface receptors could help mitigate fibrosis, inflammation, and tissue remodeling in the peritoneal cavity, thereby reducing the formation of adhesions between peritoneal surfaces [93,105,106,107,108].

Multiple stimuli regulate its expression, including growth factors, cytokines, and mechanical stress. PDGF has been implicated in peritoneal adhesion formation following surgery or injury. It stimulates the proliferation and migration of fibroblasts and myofibroblasts, leading to the deposition of extracellular matrix proteins and the formation of fibrous adhesions between peritoneal surfaces [109,110].

Additionally, PDGF stimulates angiogenesis, which may contribute to the development and persistence of peritoneal adhesions. Targeting PDGF signaling pathways represents a potential therapeutic strategy for preventing or reducing peritoneal adhesion formation. Approaches aimed at inhibiting PDGF activity or blocking its receptors could help mitigate fibrosis, angiogenesis, and tissue remodeling in the peritoneal cavity, thereby reducing the formation of adhesions between peritoneal surfaces [111].

VEGF is a homodimeric glycoprotein consisting of multiple isoforms, including VEGF-A, VEGF-B, VEGF-C, VEGF-D, and placental growth factor (PlGF). Each isoform is generated through the alternative splicing of the VEGF gene and exhibits distinct biological activities. Various cell types produce VEGF, including endothelial cells, macrophages, fibroblasts, smooth muscle cells, and tumor cells. Multiple stimuli induce its expression, including hypoxia, growth factors, cytokines, and mechanical stress. VEGF exerts its effects primarily by binding to and activating VEGF receptors (VEGFRs) expressed on the surface of endothelial cells. VEGFR activation triggers intracellular signaling pathways involved in endothelial cell proliferation, migration, survival, and vascular permeability. These processes are essential for angiogenesis, vasculogenesis, and vascular remodeling during development, wound healing, and pathological conditions, such as cancer and ischemic diseases. VEGF has been implicated in peritoneal adhesion formation following surgery or injury. It promotes angiogenesis within the peritoneal cavity, forming new blood vessels that supply nutrients and oxygen to the developing adhesions [112,113,114].

Additionally, VEGF may enhance vascular permeability and inflammatory responses, contributing to tissue edema and fibrosis associated with adhesion formation. Targeting VEGF signaling pathways represents a potential therapeutic strategy for preventing or reducing peritoneal adhesion formation. Approaches aimed at inhibiting VEGF expression or blocking its receptors could help mitigate angiogenesis, inflammation, and tissue remodeling in the peritoneal cavity, thereby reducing the formation and severity of adhesions between peritoneal surfaces [115,116].

TIMPs are small proteins typically composed of around 200 amino acids. They contain a conserved N-terminal domain responsible for binding to the active site of MMPs, inhibiting their proteolytic activity. TIMPs also possess a C-terminal domain that mediates interactions with other proteins and ECM components. Various cell types, including fibroblasts, endothelial cells, smooth muscle cells, and immune cells, produce TIMPs. Multiple stimuli, including growth factors, cytokines, and mechanical stress, regulate their expression. TIMPs regulate ECM homeostasis by inhibiting the activity of MMPs, which are responsible for degrading ECM components, such as collagen, elastin, and proteoglycans. By inhibiting MMPs, TIMPs help maintain the structural integrity of tissues and prevent excessive ECM degradation [47,117,118].

Additionally, TIMPs have been shown to modulate cell proliferation, migration, and survival through MMP-independent mechanisms. TIMPs have been implicated in peritoneal adhesion formation following surgery or injury. The dysregulation of TIMP expression and MMP/TIMP imbalance can disrupt ECM remodeling processes, leading to aberrant tissue repair and fibrosis. Both inadequate and excessive TIMP activity can contribute to pathological conditions associated with peritoneal adhesions, highlighting the importance of maintaining proper MMP/TIMP balance. Modulating TIMP expression or activity represents a potential therapeutic strategy for preventing or reducing peritoneal adhesion formation. Approaches to restoring MMP/TIMP balance could help mitigate excessive ECM remodeling, fibrosis, and tissue adhesion in the peritoneal cavity, thereby improving surgical outcomes and patient recovery [119,120].

Fibroblast Growth Factor 2 (FGF2), or essential fibroblast growth factor (bFGF), is a member of the fibroblast growth factor family. It plays diverse roles in various biological processes, including cell proliferation, differentiation, migration, and angiogenesis. GF2 is a small, secreted protein consisting of around 155 amino acids. It contains a conserved core region responsible for binding to fibroblast growth factor receptors (FGFRs) and heparan sulfate proteoglycans (HSPGs) on the cell surface. FGF2 exists in several isoforms generated by alternative splicing, with the most common forms being low-molecular-weight (LMW) and high-molecular-weight (HMW) variants. Various cell types, including fibroblasts, endothelial cells, smooth muscle cells, and tumor cells, produce FGF2. Multiple stimuli regulate its expression, including growth factors, cytokines, and mechanical stress.

FGF2 exerts its effects by binding to and activating FGFRs, receptor tyrosine kinases expressed on the surface of target cells. Upon ligand binding, FGFRs undergo dimerization and autophosphorylation, activating downstream signaling pathways, such as the Ras-MAPK and PI3K-Akt pathways. These pathways regulate cellular processes, including cell proliferation, survival, differentiation, and migration. FGF2 also stimulates angiogenesis, forming new blood vessels by promoting endothelial cell proliferation and migration. FGF2 has been implicated in peritoneal adhesion formation following surgery or injury. It promotes the proliferation and migration of fibroblasts, endothelial cells, and other cell types involved in tissue repair and remodeling [121,122,123].

Additionally, FGF2 stimulates angiogenesis, which may contribute to the development and persistence of peritoneal adhesions. Targeting FGF2 signaling pathways represents a potential therapeutic strategy for preventing or reducing peritoneal adhesion formation. Approaches aimed at inhibiting FGF2 activity or blocking its receptors could help mitigate fibrosis, angiogenesis, and tissue remodeling in the peritoneal cavity, thereby reducing the formation of adhesions between peritoneal surfaces. In summary, exploring strategies to modulate FGF2 signaling pathways may hold promise for managing peritoneal adhesions and improving patient outcomes [124,125].

In conclusion, peritoneal adhesion formation is a multifaceted process involving diverse molecular pathways and regulatory mechanisms, including integrin-mediated signaling, the regulation of E-cadherin expression and function, selectin-dependent leukocyte-endothelial interactions, and immunoglobulin superfamily molecules, such as ICAM-1 and VCAM-1, which facilitate leukocyte recruitment during inflammatory responses. Hyaluronan (HA), a critical extracellular matrix component, modulates inflammatory, fibrotic, and immune responses, providing a protective barrier that can mitigate adhesion formation. Matrix metalloproteinases (MMPs), regulated by tissue inhibitors of metalloproteinases (TIMPs), play a crucial role in extracellular matrix remodeling and turnover, influencing fibrosis and adhesion severity. The coagulation cascade, especially fibrinogen and its conversion to fibrin, along with the delicate fibrinolytic balance between tissue plasminogen activator (tPA) and plasminogen activator inhibitor-1 (PAI-1), further governs the extent and persistence of adhesions. Key growth factors, such as transforming growth factor-beta (TGF-β), vascular endothelial growth factor (VEGF), and platelet-derived growth factor (PDGF), along with interleukins and fibroblast growth factors (FGFs), actively participate in promoting angiogenesis, inflammation, and fibroblast proliferation, significantly influencing adhesion development and severity. Furthermore, Toll-like receptor (TLR)-mediated innate immune activation and tenascin-C expression critically shape the inflammatory environment, perpetuating fibrotic processes. Given the complexity and interconnectivity of these molecular mechanisms, therapeutic strategies that simultaneously or selectively target these pathways hold promise for effectively reducing or preventing postoperative adhesions. A deeper understanding of these interactions, combined with targeted molecular therapies, could significantly enhance clinical outcomes, ultimately reducing adhesion-related complications and improving patient quality of life following abdominal surgery or peritoneal injuries.

## 2. Current Limitations in Understanding Adhesion Formation

### 2.1. Single Pathway and Single Model Study Limiatation

Despite substantial advancements in molecular biology and genomics, our understanding of peritoneal adhesion formation remains incomplete due to inherent complexities in the underlying biological processes. A significant challenge in adhesion research arises from the historical reliance on investigating individual signaling pathways or isolated molecular events in simplified experimental settings. While such approaches have successfully identified numerous adhesion-related factors and potential therapeutic targets, they frequently overlook the intricate interplay and crosstalk among multiple molecular pathways simultaneously involved in adhesion formation. This reductionist methodology, typically confined to single-model studies—often employing specific animal models or in vitro systems—fails to replicate the complete biological complexity and heterogeneity observed in clinical scenarios. Consequently, translating these findings into universally effective clinical interventions has proven difficult. Addressing these limitations requires a shift toward comprehensive, integrative approaches that embrace multi-pathway analyses, cross-model comparisons, and advanced multi-omics techniques to achieve a more accurate and clinically relevant understanding of adhesion biology.

Figure 2 represents the fibrinogen pathway implicated in the peritoneal adhesions.

The following genes are implicated in the peritoneal adhesion process (Table 3).

The transforming growth factor-beta (TGF-β) gene is expressed in various cell types within the peritoneum, including mesothelial cells, fibroblasts, and immune cells. Its expression is upregulated following peritoneal injury or surgery, contributing to the pathogenesis of adhesion formation. TGF-β exerts pro-inflammatory effects by promoting the recruitment and activation of immune cells, such as macrophages and T lymphocytes, in the peritoneal cavity. Inflammation is a critical component of the adhesion formation process, and TGF-β contributes to stimulating angiogenesis, the formation of new blood vessels, in the peritoneal tissues. Angiogenesis is associated with the development and persistence of adhesions, as it facilitates the influx of nutrients, oxygen, and inflammatory cells to the injured site, promoting tissue repair and remodeling. To the inflammatory milieu within the peritoneum following injury or surgery. TGF-β regulates cell migration, proliferation, and survival in the peritoneum, thereby influencing the dynamics of tissue repair and adhesion formation. It promotes the migration of fibroblasts, mesothelial cells, and other cell types involved in tissue remodeling and wound healing processes. Targeting TGF-β signaling pathways represents a potential therapeutic strategy for preventing or reducing peritoneal adhesion formation. Approaches aimed at inhibiting TGF-β activity or blocking its receptors could help mitigate fibrosis, inflammation, and angiogenesis in the peritoneal cavity, thereby reducing the formation and severity of adhesions [88,136,137].

Vascular endothelial growth factor (VEGF) genes are expressed in various cell types within the peritoneum, including mesothelial cells, fibroblasts, and inflammatory cells. Its expression is upregulated following peritoneal injury or surgery, contributing to the pathogenesis of adhesion formation. VEGF is a potent inducer of angiogenesis, promoting the formation of new blood vessels within the peritoneal tissues. Angiogenesis facilitates the influx of nutrients, oxygen, and inflammatory cells to the injured site, promoting tissue repair and remodeling. However, it can also contribute to the development and persistence of adhesions. VEGF can stimulate inflammatory responses in the peritoneum by facilitating the recruitment and activation of immune cells, such as macrophages and neutrophils. Inflammation is a crucial component of adhesion formation, and VEGF-mediated inflammatory signaling may exacerbate adhesion formation. VEGF can regulate the migration, proliferation, and survival of various cell types in the peritoneum, including endothelial cells, fibroblasts, and mesothelial cells. These cellular responses influence the dynamics of tissue repair and adhesion formation in response to injury or surgery. Targeting VEGF signaling pathways represents a potential therapeutic strategy for preventing or reducing peritoneal adhesion formation. Approaches aimed at inhibiting VEGF activity or blocking its receptors could help mitigate angiogenesis, inflammation, and fibrosis in the peritoneal cavity, thereby reducing the formation and severity of adhesions [88,128,138].

The platelet-derived growth factor (PDGF) gene is expressed in various cell types within the peritoneum, including mesothelial cells, fibroblasts, and immune cells. Its expression is upregulated in response to tissue injury or surgery, contributing to the pathogenesis of adhesion formation. PDGF is critical in promoting fibroblast activation and proliferation in the peritoneal cavity. It acts as a potent mitogen for fibroblasts, stimulating their proliferation and migration to the site of injury, where they contribute to the formation of fibrous adhesions. This results in the accumulation of fibrous tissue at the site of injury, leading to adhesion formation, which stimulates angiogenesis and forms new blood vessels in the peritoneal tissues. Angiogenesis facilitates the influx of nutrients, oxygen, and inflammatory cells to the injured site, promoting tissue repair and remodeling, but it also contributes to the development and persistence of adhesions. PDGF can modulate inflammatory responses in the peritoneum by facilitating the recruitment and activation of immune cells, such as macrophages and neutrophils. Inflammation is a vital component of the adhesion formation process, and PDGF may exacerbate this process by enhancing the inflammatory milieu within the peritoneal cavity. Targeting PDGF signaling pathways represents a potential therapeutic strategy for preventing or reducing peritoneal adhesion formation. Approaches aimed at inhibiting PDGF activity or blocking its receptors could help mitigate fibrosis, inflammation, and angiogenesis in the peritoneal cavity, thereby reducing the formation and severity of adhesions [28,120,139].

Fibroblast growth factor (FGF) genes are expressed in various cell types within the peritoneum, including mesothelial cells, fibroblasts, and immune cells. Their expression is upregulated in response to tissue injury or surgery, contributing to the pathogenesis of adhesion formation. FGFs are vital in promoting fibroblast activation and proliferation in the peritoneal cavity. This results in the accumulation of fibrous tissue at the site of injury, leading to adhesion formation. FGFs stimulate angiogenesis, forming new blood vessels in the peritoneal tissues. Angiogenesis facilitates the influx of nutrients, oxygen, and inflammatory cells to the injured site, promoting tissue repair and remodeling, but it also contributes to the development and persistence of adhesions. FGFs can modulate inflammatory responses in the peritoneum by facilitating the recruitment and activation of immune cells, such as macrophages and neutrophils. Inflammation is a critical component of the adhesion formation process, and FGFs may exacerbate this process by enhancing the inflammatory milieu within the peritoneal cavity. Targeting FGF signaling pathways represents a potential therapeutic strategy for preventing or reducing peritoneal adhesion formation. Approaches aimed at inhibiting FGF activity or blocking its receptors could help mitigate fibrosis, inflammation, and angiogenesis in the peritoneal cavity, thereby reducing the formation and severity of adhesions [115,140].

The tissue plasminogen activator (tPA) gene, or PLAT, encodes the tissue-type plasminogen activator (tPA) protein. However, specific studies focusing on the role of the PLAT gene in peritoneal adhesions may be limited. The LAT gene could be implicated in peritoneal adhesions. The PLAT gene may be upregulated or downregulated in response to peritoneal injury or surgery, affecting the tissue-type plasminogen activator (tPA) levels in the peritoneum. Alterations in the PLAT gene expression could lead to the dysregulation of fibrinolysis, the process responsible for breaking down fibrin clots. Reduced tPA levels due to decreased PLAT gene expression may result in impaired fibrin clot dissolution and increased fibrin accumulation, contributing to adhesion formation. Genetic variants within the PLAT gene or its regulatory regions may influence individual susceptibility to peritoneal adhesions. Polymorphisms affecting PLAT gene expression or tPA activity could modulate the risk of adhesion formation following peritoneal injury. Targeting the PLAT gene or its downstream signaling pathways could represent a potential therapeutic strategy for preventing or reducing peritoneal adhesions. Strategies to restore normal tPA levels or enhance fibrinolysis could help mitigate adhesion formation and promote peritoneal tissue repair [105,116,141,142].

The plasminogen activator inhibitor-1 (PAI-1) gene, also known as SERPINE1 (serpin family E member 1), encodes the plasminogen activator inhibitor-1 protein, which is a vital regulator of the fibrinolytic system. The SERPINE1 gene consists of multiple exons and introns that undergo transcription and alternative splicing to generate different plasminogen activator inhibitor-1 (PAI-1) isoforms. Plasminogen activator inhibitor-1 (PAI-1) is a serine protease inhibitor belonging to the serpin superfamily. It inhibits the activity of tissue-type plasminogen activator (tPA) and urokinase-type plasminogen activator (uPA), thereby regulating fibrinolysis. PAI-1 plays a crucial role in regulating fibrinolysis by inhibiting the conversion of plasminogen to plasmin, which is responsible for fibrin clot dissolution.

Elevated levels of PAI-1 can lead to impaired fibrinolysis and increased fibrin accumulation, contributing to the development of peritoneal adhesions. The expression of the SERPINE1 gene can be modulated by various factors, including cytokines, growth factors, and hormones, in response to tissue injury, inflammation, or metabolic changes. The dysregulation of PAI-1 expression or activity may predispose individuals to developing peritoneal adhesions. Increased PAI-1 levels can impair fibrinolysis and promote fibrin accumulation, leading to fibrous adhesions following peritoneal injury or surgery. Elevated PAI-1 levels have been associated with various pathological conditions, including thrombosis, cardiovascular disease, and fibrotic disorders. Strategies aimed at modulating PAI-1 activity or expression may have therapeutic potential for preventing or reducing the formation of peritoneal adhesions. Targeting PAI-1 signaling pathways represents a potential therapeutic strategy for mitigating peritoneal adhesion formation. Approaches aimed at inhibiting PAI-1 activity or blocking its interactions with plasminogen activators could help restore fibrinolysis and prevent excessive fibrin accumulation in the peritoneal cavity [41,143,144].

They comprise α and β subunits and play crucial roles in cellular processes, including cell adhesion, migration, signaling, and differentiation. Integrin genes are located on different chromosomes in humans, with each α or β subunit having its specific genomic location. Integrin genes consist of multiple exons and introns that encode the α and β subunits of the integrin receptors. Integrin receptors are heterodimers composed of α and β subunits containing extracellular, transmembrane, and cytoplasmic domains. The dysregulation of integrin expression or activity may contribute to the pathogenesis of peritoneal adhesions and related disorders. Targeting integrin signaling pathways represents a potential therapeutic strategy for preventing or reducing adhesion formation. Strategies to modulate integrin expression, activation, or function could help mitigate peritoneal adhesion formation and improve clinical outcomes in patients undergoing abdominal surgery or experiencing peritoneal injury [10,88,145].

The dysregulation of selectin expression or activity has been implicated in various inflammatory and immune-related disorders, including peritoneal adhesions, inflammatory bowel disease, and atherosclerosis. Modulating selectin–ligand interactions or selectin expression could represent a potential therapeutic strategy for preventing or reducing peritoneal adhesion formation and attenuating inflammatory responses in the peritoneal cavity [146,147].

IgSF molecule genes contain multiple exons and introns that encode the IgSF proteins. The extracellular domains of IgSF molecules typically include one or more immunoglobulin-like domains responsible for cell adhesion and protein–protein interactions. IgSF molecules are characterized by immunoglobulin-like domains, which mediate homophilic or heterophilic interactions between cells or between cells and extracellular matrix components. IgSF molecules play crucial roles in cell adhesion, migration, signaling, and immune responses. They mediate cell–cell and cell–matrix interactions by binding to specific ligands or counter-receptors expressed on neighboring cells or the extracellular matrix. IgSF molecules are involved in the regulation of leukocyte trafficking, inflammatory responses, and tissue repair processes associated with peritoneal adhesion formation. They facilitate the adhesion, transmigration, and activation of leukocytes within the peritoneal cavity during inflammatory and immune responses. IgSF molecules regulate various cellular behaviors implicated in peritoneal adhesion formation, including the cell adhesion, migration, and activation of immune cells. IgSF molecule expression or activity dysregulation has been involved in various inflammatory and autoimmune diseases, cancer metastasis, and neurodevelopmental disorders. Targeting IgSF molecule-mediated cell adhesion and signaling pathways represents a potential therapeutic strategy for modulating inflammatory responses and mitigating peritoneal adhesion formation. Approaches aimed at blocking IgSF molecule-ligand interactions or modulating IgSF molecule expression could help reduce leukocyte recruitment and inflammation in the peritoneum [148,149,150]

The proteolytic cleavage of the prodomain activates MMPs, which are synthesised as zymogens (pro-MMPs). In addition to other domains involved in substrate binding and regulation, the active forms of MMPs have catalytic domains that are in charge of breaking down ECM. By breaking down different components of the extracellular matrix (ECM), such as collagen, glycoproteins, and proteoglycans, MMPs play important roles in tissue remodeling, wound healing, inflammation, angiogenesis, and cancer progression. During tissue repair processes linked to the production of peritoneal adhesions, MMPs play a role in remodeling the peritoneal extracellular matrix. Excessive ECM degradation or deposition can result in aberrant tissue remodeling and adhesion creation. One possible cause of this is the dysregulation of MMP expression or activity. Tissue inhibitors of metalloproteinases (TIMPs), growth hormones, cytokines, mechanical stimuli, and other variables can all control the production and activity of MMPs. Multiple illnesses, including cancer and fibrosis, have been linked to the imbalance between MMPs and TIMPs. The metastasis of cancer and other pathological illnesses, like fibrosis, arthritis, and cardiovascular diseases, have all been linked to the dysregulated expression or activity of MMPs. One possible therapeutic approach to prevent or lessen the formation of peritoneal adhesions is to modify MMP expression or activity. Targeting MMP activity or control could be one way to influence adhesion formation and ECM remodeling [151,152,153].

Endogenous inhibitors known as the tissue inhibitors of metalloproteinases (TIMPs) control the activity of MMPs and other metalloproteinases. Four identified TIMP genes in humans encode proteins that regulate MMP activity and uphold the equilibrium between ECM synthesis and breakdown. In humans, TIMP genes are distributed over several chromosomes, with a distinct chromosomal position assigned to each TIMP subtype. The TIMP proteins are encoded by the many exons and introns that make up TIMP genes. These proteins bind to the active site of MMPs and stop their enzymatic activity because of a conserved cysteine-rich domain. TIMPs are tiny glycoproteins with a 1:1 stoichiometric ratio of reversible binding to the catalytic domain of MMPs. This binding controls ECM turnover and stops MMPs from destroying ECM components. By regulating MMP activity and preserving the equilibrium between ECM production and degradation, TIMPs are essential for tissue remodeling, wound healing, inflammation, and the advancement of cancer. TIMPs play a role in controlling the ECM remodeling processes that lead to the formation of peritoneal adhesions. The imbalance between TIMPs and MMPs may disrupt ECM homeostasis and contribute to abnormal tissue remodeling and adhesion formation. Various factors, including growth factors, cytokines, mechanical stimuli, and ECM components, can regulate the expression and activity of TIMPs. Alterations in TIMP expression or activity have been implicated in the pathogenesis of various diseases, including fibrosis and cancer. The dysregulated expression or activity of TIMPs has been associated with pathological conditions, such as fibrosis, arthritis, cardiovascular diseases, and cancer metastasis. Modulating TIMP expression or activity represents a potential therapeutic strategy for preventing or reducing peritoneal adhesion formation. Approaches to restoring the balance between TIMPs and MMPs could help regulate ECM remodeling and favor peritoneal adhesion [154,155,156].

By identifying common chemical patterns linked to infections, the family of pattern recognition receptors (PRRs), known as Toll-like receptors (TLRs), plays a vital role in the innate immune system. A collection of genes encodes TLRs in humans. There are distinct chromosomes on which TLR genes are found, and each subtype of TLR has a unique genomic location. The TLR proteins are encoded by various exons and introns found in TLR genes. Type I transmembrane proteins called TLRs are distinguished by cytoplasmic Toll/IL-1 receptor (TIR) domains that are involved in signaling and extracellular leucine-rich repeat (LRR) domains that are involved in ligand recognition. TLRs are membrane-bound receptors that have two domains, an intracellular one for signal transduction and an extracellular one for ligand recognition. The receptor undergoes conformational changes upon ligand binding, which activate downstream signaling pathways. TLRs are able to identify many components of microorganisms, including lipopolysaccharides, lipoproteins, nucleic acids, and proteins that come from parasites, bacteria, viruses, and fungi. Upon ligand interaction, TLRs start signaling cascades that activate antimicrobial defense systems and inflammatory responses. TLRs identify endogenous danger signals emitted during peritoneal inflammation or tissue damage, as well as microbiological infections. The start and regulation of inflammatory reactions linked to the creation of peritoneal adhesions are facilitated by the activation of TLR signaling pathways. Cellular stress, cytokines, and microbial products are a few of the variables that can control TLR expression and function. Numerous inflammatory and autoimmune disorders have been linked to TLR signaling dysregulation as an etiology. Aberrant TLR activation or signaling has been associated with pathological conditions, such as sepsis, inflammatory bowel disease, rheumatoid arthritis, and cancer. Modulating TLR activation or signaling represents a potential therapeutic strategy for regulating inflammatory responses and mitigating peritoneal adhesion formation. Approaches aimed at targeting TLR expression, ligand binding, or downstream signaling pathways could help modulate the peritoneum’s immune response and tissue repair processes [79,157,158].

Tenascin-C is an extracellular matrix glycoprotein involved in various physiological and pathological processes, including tissue development, wound healing, inflammation, and cancer progression. In humans, the gene encoding tenascin-C is called TNC. The TNC gene is located on chromosome 9q33.1 in humans. The TNC gene contains multiple exons and introns that encode the tenascin-C protein. The alternative splicing of the TNC pre-mRNA generates different isoforms of tenascin-C with distinct domain structures and functional properties. Tenascin-C is expressed at sites of tissue injury, inflammation, and fibrosis in the peritoneum and contributes to the formation of adhesion tissues by promoting cell migration, matrix remodeling, and fibrogenesis. The increased expression of tenascin-C has been observed in peritoneal adhesions and is associated with tissue fibrosis and impaired wound healing. Various factors, including growth factors, cytokines, mechanical stimuli, and tissue injury, can regulate the expression of the TNC gene. The increased expression of tenascin-C has been linked to the development of a number of illnesses, such as inflammatory conditions, cancer, and cardiovascular diseases. Pathological disorders, including cancer metastasis, autoimmune illnesses, tissue fibrosis, and chronic inflammation, have all been linked to the dysregulated expression or function of tenascin-C. One possible treatment approach for avoiding or decreasing the production of peritoneal adhesions is to modify the expression or activity of tenascin-C. Strategies that target the tenascin-C signaling pathways or engage with cell surface receptors may be able to regulate the processes involved in tissue repair and adhesion creation [159,160,161].

Adhesion formation is represented in Figure 1. Increased levels of TGF-ß2, IRF-1, IFN-γ1, TGF-ß1, ß1, VEGF, and H2O2 are all implicated in peritoneal adhesion formation.

The adhesion formation mechanism involves TGF-ß2, IRF-1, IFN-γ1, TGF-ß1, ß1, VEGF, and H2O2. HIF-1α—hypoxia-inducible factor 1-alpha; NF-kB—nuclear factor kappa-light-chain-enhancer of activated B cells; TGF-ß2—transforming growth factor-beta 2; IRF-1—interferon regulatory factor 1; TNF-α—tumor necrosis factor alpha; IFN-γ1—interferon gamma; TGF-ß1—transforming growth factor beta 1; VEGF—vascular endothelial growth factor; H2O2—hydrogen peroxide; IL-1,2,4,6,10,17—interleukin-1,2,4,6,10,17; NO—nitric oxide; OONO—peroxynitrite; P53—tumor protein P53; COX-2—cyclooxygenase-2;TIMP-1,2—metallopeptidase inhibitor 1,2. As seen, the tissue injury during surgery results in devascularization and, consequently, in tissue hypoxia. Those changes stimulate the anaerobic metabolism that increases the oxidative stress, resulting in increased levels of HIF-1α and NF-kB. An increased level of HIF-1α and NF-kB stimulates the TGF-ß2, IRF-1, IFN-γ1, TGF-ß1, ß1, VEGF, and H2O2 levels. Increasing levels of TGF-ß2 result in decreased levels of ↓IL-2 and an increase in IL-1, 4, 6, 10, 17.

Furthermore, increased levels of IRF-1 and TNF-α produce NO and OONO, leading to increased protein nitration. Also, IFN-γ1 increases COX-1 levels, which leads to cell proliferation. VEGF stimulates TIMP-1,2, HIF-1α, and NF-kB. All of these processes lead to adhesion development.

### 2.2. Complexity of Biological Networks in Adhesion Pathology

While significant advances have been achieved in understanding the molecular characterization of peritoneal adhesion formation, several critical limitations remain. Current knowledge primarily derives from isolated pathway analyses or single-model studies, which inadequately capture the complex, interconnected biological networks driving adhesion pathology. Furthermore, despite the expanding availability of genomic, transcriptomic, proteomic, and epigenomic data, there is limited synthesis and systematic integration of these datasets to elucidate comprehensive molecular mechanisms [162].

Future research efforts should prioritize adopting robust, integrative multi-omics strategies, combining genomic, transcriptomic, proteomic, metabolomic, and epigenomic analyses. Such an integrative approach would provide deeper insights into adhesion biology by capturing molecular interactions at multiple biological layers. High-throughput sequencing technologies, single-cell RNA sequencing, mass spectrometry-based proteomics, and advanced metabolomics profiling will be instrumental in these endeavors [163].

Moreover, studies must rigorously incorporate advanced bioinformatics and systems biology methodologies. Tools such as network biology modeling, pathway enrichment analyses, and the computational prediction of protein–protein interactions (e.g., via STRING, Cytoscape) will enable researchers to delineate precise molecular interactions and regulatory networks comprehensively. Leveraging databases, such as the Gene Expression Omnibus (GEO), The Cancer Genome Atlas (TCGA), and bioinformatics resources like KEGG and Reactome, can facilitate the systematic exploration and validation of adhesion-related pathways [65].

Machine learning and artificial intelligence algorithms could significantly enhance the discovery and validation of robust predictive biomarkers and novel therapeutic targets. These predictive modeling approaches may identify patient-specific susceptibility markers and stratify patients based on their genetic and molecular risk profiles, potentially paving the way for personalized therapeutic strategies.

Furthermore, expanding research into experimental models, including human-derived 3D tissue cultures, organoids, and animal models that closely mimic clinical scenarios, will strengthen translational research outcomes. These advanced models could improve the relevance of preclinical findings to clinical practice, accelerating the transition from bench to bedside [164].

Lastly, the clinical translation of promising molecular targets must be pursued through collaborative interdisciplinary research, bringing together surgeons, molecular biologists, bioinformaticians, pharmacologists, and industry partners. Such collaborations are essential for translating molecular insights into practical, targeted interventions and clinical therapies to mitigate adhesion formation effectively and improve patient outcomes [165].

## 3. Multi-Omics Integration in Adhesion Research

In recent years, multi-omics approaches—which encompass genomics, transcriptomics, proteomics, metabolomics, and epigenomics—have emerged as powerful methodologies for advancing our understanding of complex biological phenomena, including peritoneal adhesion formation. By integrating data across multiple molecular levels, multi-omics analyses offer a comprehensive view of the regulatory networks and interactive pathways involved in adhesion biology. This holistic strategy helps overcome the inherent limitations associated with single-pathway studies or isolated model systems, providing deeper insights into the intricate molecular interplay underlying fibrosis, inflammation, and tissue remodeling associated with adhesions. Leveraging advanced bioinformatics tools and systems biology approaches, multi-omics integration facilitates the identification of robust biomarkers, novel therapeutic targets, and personalized treatment strategies tailored to individual molecular profiles. Ultimately, these integrative approaches hold significant promise for translating molecular insights into effective clinical solutions to prevent or mitigate adhesion-related complications following surgical interventions or peritoneal injury. Several novel developments may have occurred in the field of peritoneal adhesions. Some potential areas of advancement are listed in the table below (Table 4).

Regarding the genetic studies on peritoneal adhesions, some significant studies are listed in the table below. These studies contribute to our understanding of the genetic basis of peritoneal adhesions, shedding light on potential mechanisms involved in adhesion formation and suggesting avenues for further research and therapeutic development (Table 5).

In a study regarding the genetic polymorphisms associated with adhesion formation (#1, Table 5), researchers analyzed genetic data from individuals with and without peritoneal adhesions to identify differences in genetic polymorphisms. Statistical analyses were used to assess the significance of these associations. The study sought to understand the genetic basis of adhesion formation and identify potential genetic markers for susceptibility to adhesions. Identifying genetic polymorphisms associated with adhesion formation could have significant clinical implications. It could enable healthcare providers to identify individuals at higher risk of developing adhesions and implement preventive measures or surveillance strategies accordingly. Moreover, targeting specific genetic pathways involved in adhesion formation could lead to developing novel therapeutic interventions to reduce adhesion-related complications following surgery or injury.

The objective of the research was to determine possible treatment targets and clarify the molecular mechanisms driving adhesion development. By identifying dysregulated genes during adhesion formation, researchers aimed to uncover vital biological pathways involved in this process, which could lead to targeted therapies for preventing or treating adhesions. Understanding the gene expression changes associated with adhesion formation could provide insights into the pathogenesis of this condition and identify novel therapeutic targets. By targeting specific genes or signaling pathways implicated in adhesion formation, researchers may be able to develop more effective strategies for preventing or treating adhesions, ultimately improving patient outcomes following surgery or injury.

The Association Studies of Candidate Genes (#3, Table 5) aimed to investigate the association between specific candidate genes and the risk of developing peritoneal adhesions. Candidate genes are selected based on their known or hypothesized roles in biological processes relevant to adhesion formation, such as inflammation, tissue repair, or cell adhesion. By analyzing genetic variations within these candidate genes, researchers sought to identify genetic markers or polymorphisms that may predispose individuals to developing adhesions. Researchers likely selected candidate genes based on prior knowledge of their involvement in processes related to adhesion formation. They then collected genetic data from individuals with and without peritoneal adhesions and clinical information about adhesion status. Genetic variations within candidate genes were analyzed using techniques such as genotyping or sequencing. Statistical analyses assessed the association between specific genetic variants and adhesion susceptibility. The study aimed to identify genetic markers or polymorphisms associated with an increased risk of developing peritoneal adhesions. By elucidating the genetic factors underlying adhesion formation, researchers aimed to improve our understanding of the pathogenesis of this condition and identify potential targets for preventive or therapeutic interventions. Identifying genetic associations with peritoneal adhesions could have significant clinical implications. It could enable healthcare providers to identify individuals at higher risk of developing adhesions and implement preventive measures or surveillance strategies accordingly. Moreover, understanding the genetic basis of adhesion formation could lead to the development of personalized treatment approaches tailored to an individual’s genetic profile [196,197].

The study on Genome-Wide Association Studies (GWAS) in Peritoneal Adhesions (#4, Table 4) aimed to systematically investigate the genetic basis of peritoneal adhesions using genome-wide association studies (GWAS). GWAS is a powerful approach that allows researchers to examine genetic variations across the entire genome and identify common genetic variants associated with a particular trait or disease. By analyzing genetic data from a large cohort of individuals with and without peritoneal adhesions, researchers sought to identify novel genetic loci and pathways associated with adhesion formation susceptibility. Researchers likely collected genetic data from a well-characterized cohort of individuals with peritoneal adhesions and control individuals without adhesions. Genetic variations across the genome were analyzed using high-throughput genotyping or sequencing technologies. Statistical analyses, such as logistic regression or chi-square tests, were used to identify genetic variants significantly associated with adhesion susceptibility. Genome-wide significance thresholds were applied to account for multiple comparisons. The study’s main objective was to identify novel genetic loci and pathways related to the risk of developing peritoneal adhesions. By uncovering new genetic associations, researchers aimed to improve our understanding of the biological mechanisms underlying adhesion formation and identify potential targets for preventive or therapeutic interventions. GWAS in peritoneal adhesions could provide valuable insights into the genetic determinants of this condition and lead to the identification of novel therapeutic targets. Understanding the genetic basis of adhesion formation could pave the way for developing personalized treatment approaches and improving patient outcomes following surgery or injury [198,199].

Through analyzing alterations in gene expression, protein concentration, and metabolic patterns linked to adhesion formation, scientists seek to identify crucial regulatory routes and molecular targets for potential intervention. Studies utilizing functional genomics offer a significant understanding of the molecular processes underlying the creation of peritoneal adhesions, illuminating the intricate interactions among genes, proteins, and signaling pathways. Researchers may find novel therapeutic targets for peritoneal adhesion prevention or treatment by discovering particular genes or pathways that are dysregulated in adhesion-prone tissues, ultimately improving patient outcomes. Studies using functional genomics are essential for expanding our knowledge of the molecular processes underlying peritoneal adhesions and provide encouraging avenues for the creation of tailored pharmaceuticals.

The association of Inflammatory Genes with Adhesion Formation (#7, Table 5) study investigates the association between genes involved in inflammation and the formation of peritoneal adhesions. Inflammation plays a crucial role in the pathogenesis of adhesions, as it promotes tissue injury, fibrosis, and the recruitment of immune cells to the site of injury. Genes encoding pro-inflammatory cytokines, chemokines, and other inflammatory mediators may influence the inflammatory response and contribute to adhesion formation. Researchers likely analyzed genetic variations within inflammatory genes in individuals with and without peritoneal adhesions. Genetic variants, such as single-nucleotide polymorphisms (SNPs) or copy number variations (CNVs), may affect the expression or function of inflammatory genes, thereby influencing an individual’s susceptibility to adhesion formation. Genetic data were collected using techniques such as genotyping or sequencing, and statistical analyses were used to assess the association between specific genetic variants and adhesion susceptibility. The main objective of this study was to elucidate the role of inflammatory genes in the pathogenesis of peritoneal adhesions. By identifying genetic variants associated with adhesion formation, researchers aimed to uncover critical inflammatory pathways and potential therapeutic targets for intervention. Understanding the genetic basis of inflammation in adhesion formation could lead to the development of targeted anti-inflammatory therapies to prevent or reduce adhesion formation. Identifying the association between inflammatory genes and adhesion formation is crucial for understanding the underlying mechanisms of this condition. Inflammatory pathways represent promising targets for therapeutic intervention, as anti-inflammatory agents may help mitigate the inflammatory response and prevent excessive tissue fibrosis and adhesion formation. Moreover, genetic markers associated with adhesion susceptibility could be used to identify individuals at higher risk of developing adhesions and implement personalized preventive strategies. This study contributes to our understanding of the role of inflammation in peritoneal adhesion formation and highlights the potential of targeting inflammatory pathways for the prevention and treatment of adhesions [200,201].

In growth factors and their receptors’ function in the formation of adhesion (#8, Table 5), the pathophysiology of peritoneal adhesions is examined in relation to growth factors and their receptors. Growth factors are signaling molecules that control migration, differentiation, and proliferation of cells, among other biological activities. Adhesion development can result from aberrant tissue healing and fibrosis caused by the dysregulation of growth factor signaling pathways. Growth factor receptors are essential for mediating physiological responses, because they transduce the signals that growth factors generate, such as receptor tyrosine kinases. Researchers most likely looked at the receptors, activation states, and the expression levels of growth factors in peritoneal tissues taken from people with and without adhesions. Growth factors and their receptors’ protein expression or phosphorylation levels may have been evaluated using methods like immunohistochemistry, Western blotting, or quantitative PCR. Furthermore, it is possible that functional tests were carried out to assess how growth factor signaling affects cellular functions that are important for adhesion creation, like cell division, migration, and extracellular matrix synthesis. Clarifying the function of growth factors and their receptors in mediating the pathophysiology of peritoneal adhesions was the primary goal of this investigation. By investigating the expression and activity of growth factor signaling pathways, researchers aimed to identify potential therapeutic targets for intervention. Understanding how dysregulated growth factor signaling contributes to adhesion formation could lead to the development of targeted therapies aimed at modulating growth factor activity and mitigating adhesion formation. Growth factors and their receptors represent promising targets for therapeutic intervention in peritoneal adhesions. Modulating growth factor signaling pathways may help regulate cellular processes involved in tissue repair and fibrosis, thereby reducing adhesion formation. The targeted inhibition of specific growth factor receptors or downstream signaling molecules could offer novel therapeutic strategies for preventing or treating adhesions, ultimately improving patient outcomes following surgery or injury. This study contributes to our understanding of the molecular mechanisms underlying peritoneal adhesion formation and highlights the potential of targeting growth factor signaling pathways for therapeutic intervention [202,203].

The Genetic Variation in Extracellular Matrix Proteins and Adhesion Formation (#9, Table 5) study investigates the role of genetic variation in extracellular matrix (ECM) proteins in the pathogenesis of peritoneal adhesions. The ECM is a complex network of proteins and carbohydrates that provides structural support to tissues and regulates cellular behavior. The dysregulation of ECM composition or remodeling can lead to abnormal tissue repair and fibrosis, contributing to the formation of adhesions. Genetic variations in genes encoding ECM proteins may influence ECM structure, function, and turnover, affecting an individual’s susceptibility to adhesion formation. Researchers likely examined genetic variations within genes encoding ECM proteins in individuals with and without peritoneal adhesions. Genetic variants such as single nucleotide polymorphisms (SNPs) or copy number variations (CNVs) may affect ECM proteins’ expression, structure, or function, potentially influencing adhesion formation. Genetic data were collected using techniques such as genotyping or sequencing, and statistical analyses were used to assess the association between specific genetic variants and adhesion susceptibility. The main objective of this study was to elucidate how genetic variation in ECM proteins contributes to the pathogenesis of peritoneal adhesions. By identifying genetic variants associated with adhesion formation, researchers aimed to uncover vital molecular mechanisms and potential therapeutic targets for intervention. Understanding how genetic variation influences ECM composition and remodeling could lead to the development of personalized treatment approaches for preventing or treating adhesions. Genetic variation in ECM proteins represents an essential determinant of adhesion formation susceptibility. ECM proteins play crucial roles in tissue repair, inflammation, and cell adhesion, and genetic variants affecting ECM function may contribute to abnormal tissue remodeling and fibrosis associated with adhesion formation. Targeting specific ECM proteins or pathways affected by genetic variation could offer novel therapeutic strategies for preventing or treating peritoneal adhesions, ultimately improving patient outcomes following surgery or injury. This study contributes to our understanding of the genetic determinants of peritoneal adhesion formation and highlights the potential of targeting ECM proteins [204,205].

The researchers of Epigenetic Modifications and Adhesion Formation (#10, Table 5) likely analyzed epigenetic modifications in peritoneal tissues obtained from individuals with and without adhesions. Techniques such as bisulfite sequencing, chromatin immunoprecipitation sequencing (ChIP-seq), and RNA sequencing may have been used to profile DNA methylation patterns, histone modification profiles, and non-coding RNA expression levels, respectively. Computational analyses were then employed to identify differentially methylated regions, histone modification patterns, and dysregulated non-coding RNAs associated with adhesion formation. The main objective of this study was to elucidate how epigenetic modifications contribute to the pathogenesis of peritoneal adhesions. By investigating the epigenetic landscape of adhesion-prone tissues, researchers aimed to identify fundamental regulatory mechanisms and potential therapeutic targets for intervention. Understanding how epigenetic modifications influence gene expression and cellular processes in adhesion formation could lead to the development of targeted epigenetic therapies for preventing or treating adhesions.

In adhesion Formation Affected by miRNA Dysregulation (#11, Table 5), the pathophysiology of peritoneal adhesions is examined in relation to the dysregulation of microRNAs (miRNAs). Small, non-coding RNA molecules known as miRNAs are crucial for the control of post-transcriptional gene expression. They bind to target mRNAs’ 3′ untranslated regions (UTRs), which causes translational suppression or mRNA destruction. A number of disease processes, including inflammation, fibrosis, and tissue remodeling—basic mechanisms underpinning adhesion formation—have been linked to the dysregulation of miRNA expression. MiRNA expression profiles in peritoneal tissues from people with and without adhesions were probably examined by researchers. MiRNA expression levels may have been measured via RNA sequencing or microarray analysis. After that, computational research was used to find miRNAs that were differently expressed and connected to adhesion development. The regulatory effects of dysregulated miRNAs on target genes involved in adhesion formation may have been verified by functional assays, such as luciferase reporter assays or gain- and loss-of-function studies. This study’s primary goal was to clarify the role that miRNA dysregulation plays in the pathophysiology of peritoneal adhesions. In order to identify important regulatory mechanisms and possible therapeutic targets for intervention, researchers identified dysregulated miRNAs and their target genes. MiRNA-based therapeutics for adhesion prevention or treatment may be developed as a result of a better understanding of the effects of miRNA dysregulation on gene expression and cellular processes in adhesion formation. One interesting target for therapeutic intervention in peritoneal adhesions is miRNA dysregulation. Dysregulated miRNA expression may be a factor in abnormal tissue remodeling linked to adhesion formation. miRNAs are essential in controlling gene expression networks involved in inflammation, fibrosis, and tissue repair. By focusing on dysregulated miRNAs or the genes downstream of these miRNAs, researchers may be able to develop new therapeutic approaches to treat or prevent adhesions, which would eventually improve patient outcomes after injury or surgery [119,206].

The Function of Cytokine Signaling Routes in the Formation of Adhesion (#12, Table 5) study looks into how cytokine signaling pathways contribute to the development of peritoneal adhesions. Small signaling proteins called cytokines control inflammation, tissue healing, and immunological responses. Fundamental mechanisms driving adhesion creation, such as fibrosis, abnormal tissue remodelling, and chronic inflammation, can be attributed to the dysregulation of cytokine signaling pathways. Researchers likely analyzed cytokine expression levels and signaling pathway activation in peritoneal tissues obtained from individuals with and without adhesions. Techniques such as enzyme-linked immunosorbent assays (ELISA), immunohistochemistry, or Western blotting may have been used to quantify cytokine expression levels or phosphorylation levels of signaling pathway components. Functional assays, such as cell culture models or animal studies, may have been employed to investigate the effects of cytokine stimulation or inhibition on adhesion formation in vitro or in vivo. The main objective of this study was to elucidate how cytokine signaling pathways contribute to the pathogenesis of peritoneal adhesions. By investigating the expression and activity of cytokines and their downstream signaling pathways, researchers aimed to identify potential therapeutic targets for intervention. Understanding how dysregulated cytokine signaling promotes inflammation, fibrosis, and tissue remodeling in adhesion formation could lead to the development of targeted therapies for preventing or treating adhesions. Cytokine signaling pathways represent promising targets for therapeutic intervention in peritoneal adhesions. Modulating cytokine activity or blocking specific cytokine receptors may help mitigate inflammation, fibrosis, and aberrant tissue remodeling associated with adhesion formation. The targeted inhibition of cytokine signaling pathways could offer novel therapeutic strategies for preventing or treating adhesions, ultimately improving patient outcomes following surgery or injury [207,208].

The Identification of Novel Genetic Risk Loci for Adhesion Formation (#13, Table 5) study aimed to identify novel genetic risk loci associated with the formation of peritoneal adhesions. Genetic risk loci are specific regions of the genome that are statistically associated with an increased risk of developing a particular trait or disease. By conducting genome-wide association studies (GWAS) or targeted genetic analyses, researchers sought to uncover genetic variants or loci predisposing individuals to adhesion formation. Identifying novel genetic risk loci could provide insights into the genetic basis of adhesion formation and reveal potential therapeutic targets for intervention. Researchers likely collected genetic data from a large cohort of individuals with and without peritoneal adhesions. High-throughput genotyping or sequencing technologies were used to analyze genetic variations across the entire genome or within specific genomic regions of interest. Statistical analyses, such as logistic regression or genome-wide association analysis, were employed to identify genetic variants or loci significantly associated with adhesion susceptibility. Functional studies may have been conducted to investigate the biological relevance of identified genetic risk loci in adhesion formation. The primary goal of this study was to identify new genetic risk loci that are involved in the formation of peritoneal adhesions. Researchers want to gain a better knowledge of the genetic factors influencing adhesion development and find new targets for therapeutic or preventive interventions by discovering novel genetic correlations. Comprehending the genetic foundation of adhesion formation may facilitate the creation of customized treatment strategies based on an individual’s genetic makeup. Understanding the underlying genetic underpinnings of this illness depends on the identification of novel genetic risk loci for adhesion development. Genes or biological pathways implicated in adhesion formation may be identified by genetic risk loci, offering insights into the molecular mechanisms underlying this intricate trait. Additionally, genetic risk loci may function, as biomarkers to identify people who are more likely to develop adhesions and provide information for individualized treatment plans or preventative measures. This work emphasizes the significance of genetic variables in the etiology of this illness and marks a significant advancement in our understanding of the genetic determinants of peritoneal adhesion formation [209,210].

In the Adhesion Formation and MicroRNA (miRNA) Profiling (#14-Table 5) study, the objective of this research was to examine the variations in the expression of microRNAs, or miRNAs, in the peritoneal tissues of subjects who had adhesion formation and those who did not. Small, non-coding RNA molecules known as miRNAs attach to target messenger RNAs (mRNAs) and alter their expression, which is how they play crucial roles in post-transcriptional gene regulation. Dysregulated miRNA expression has been implicated in various pathological conditions, including fibrosis and tissue remodeling, which are central processes in adhesion formation. The researchers collected peritoneal tissue samples from patients undergoing surgery, with some having a history of adhesion formation and others serving as controls. Total RNA, including miRNAs, was extracted from the tissue samples and subjected to miRNA expression profiling using high-throughput techniques, such as microarrays or next-generation sequencing (NGS). Bioinformatics analyses were then performed to identify differentially expressed miRNAs between the adhesion and control groups. The main objective of this study was to identify dysregulated miRNAs associated with adhesion formation and gain insights into their potential roles in the pathogenesis of this condition. By comparing miRNA expression profiles between adhesion-prone and non-adhesion-prone tissues, the researchers aimed to uncover miRNAs that may serve as diagnostic biomarkers or therapeutic targets for adhesion-related disorders. Understanding the regulatory roles of dysregulated miRNAs in adhesion formation could lead to developing novel strategies for prevention and treatment. MiRNA profiling in adhesion formation provides valuable insights into the molecular mechanisms underlying this condition. Dysregulated miRNAs identified through profiling analyses may represent potential biomarkers for adhesion risk assessment or treatment response monitoring. Moreover, miRNAs implicated in adhesion formation could serve as targets for therapeutic intervention. Modulating the expression or activity of dysregulated miRNAs may offer new avenues for developing miRNA-based therapies to prevent or resolve peritoneal adhesions [211,212].

The Genetic Variation in Cell Adhesion Molecules and Adhesion Formation (#15, Table 5) study aimed to investigate the role of genetic variation in cell adhesion molecules (CAMs) in the pathogenesis of peritoneal adhesions. CAMs are cell surface proteins mediating cell–cell and cell–extracellular matrix interactions, playing essential roles in cell migration, tissue repair, and immune responses. Genetic variations within CAM genes may influence their expression, structure, or function, affecting cellular adhesion properties and contributing to adhesion formation. The researchers collected genetic data from individuals with and without peritoneal adhesions, focusing on variations within genes encoding CAMs. High-throughput genotyping or sequencing techniques were used to analyze genetic variants within CAM genes, such as single-nucleotide polymorphisms (SNPs) or copy number variations (CNVs). Statistical analyses were then performed to assess the association between specific genetic variants and adhesion susceptibility, considering potential confounding factors. The main objective of this study was to elucidate how genetic variation in CAMs contributes to the pathogenesis of peritoneal adhesions. By identifying genetic variants associated with adhesion formation, the researchers aimed to uncover vital molecular mechanisms and potential therapeutic targets for intervention. Understanding how genetic variation influences CAM function and cellular adhesion processes could provide insights into the pathophysiology of adhesion formation and inform the development of personalized treatment approaches. Genetic variation in CAMs represents an essential determinant of adhesion formation susceptibility. CAMs play critical roles in mediating cell–cell and cell–matrix interactions during tissue repair and remodeling, and genetic variants affecting CAM function may contribute to aberrant adhesion formation. Identifying genetic variants associated with adhesion susceptibility could lead to developing biomarkers for risk assessment and personalized preventive strategies for individuals predisposed to adhesions. Moreover, CAM-targeted therapies could be explored to prevent or treat peritoneal adhesions, potentially improving patient outcomes following surgery or injury. This study contributes to our understanding of the genetic determinants of peritoneal adhesion formation and highlights the potential of targeting cell adhesion molecules for therapeutic intervention in this condition [213,214].

Moreover, genes implicated in adhesion formation could serve as targets for therapeutic intervention. Modulating the expression or activity of genes involved in crucial biological processes underlying adhesion formation may offer new avenues for developing targeted therapies to prevent or resolve peritoneal adhesions.

The Impact of Genetic Polymorphisms on Inflammatory Signaling Pathways (#17, Table 5) study aimed to investigate the effects of genetic polymorphisms on inflammatory signaling pathways involved in the pathogenesis of peritoneal adhesions. Genetic polymorphisms are variations in the DNA sequence that can affect gene expression, protein function, and cellular signaling pathways. Inflammation plays a crucial role in adhesion formation, and genetic variations within genes encoding components of inflammatory signaling pathways may influence an individual’s susceptibility to adhesions. The researchers collected genetic data from individuals with and without peritoneal adhesions, focusing on polymorphisms within genes involved in inflammatory signaling pathways. High-throughput genotyping techniques were used to analyze genetic variants within candidate genes, such as single-nucleotide polymorphisms (SNPs). The relationship between genetic polymorphisms and adhesion susceptibility was then examined statistically, taking into account confounding variables including age, sex, and comorbidities. This study’s primary goal was to clarify the role that genetic polymorphisms in inflammatory signaling pathways play in the pathophysiology of peritoneal adhesions. The researchers sought to reveal critical molecular pathways underpinning inflammation-driven adhesion development and find possible therapeutic targets for intervention by identifying genetic variations linked to adhesion production. Understanding how genetic polymorphisms modulate inflammatory responses could provide insights into personalized risk assessment and treatment strategies for individuals predisposed to adhesions. Genetic polymorphisms in inflammatory signaling pathways represent essential determinants of adhesion formation susceptibility. Inflammation plays a central role in the pathogenesis of peritoneal adhesions, and genetic variations within genes involved in inflammatory responses may influence an individual’s risk of developing adhesions. Identifying genetic polymorphisms associated with adhesion susceptibility could lead to developing biomarkers for risk assessment and personalized preventive strategies for individuals at a higher risk of adhesion. Moreover, targeting inflammatory signaling pathways may offer novel therapeutic approaches for preventing or treating peritoneal adhesions. This study contributes to our understanding of the genetic determinants of peritoneal adhesion formation by investigating the impact of genetic polymorphisms on inflammatory signaling pathways [133,215].

The purpose of the Genetic Predisposition to Adhesion Formation and Oxidative Stress (#18, Table 5) study was to look into how oxidative stress susceptibility at the genetic level affects the pathophysiology of peritoneal adhesions. Damage to cells and inflammation result from oxidative stress, which is caused by an imbalance between the body’s antioxidant defenses and the generation of reactive oxygen species (ROS). Through the modulation of oxidative stress levels and inflammatory responses, genetic differences in genes encoding antioxidant enzymes and regulators of oxidative stress response pathways may affect an individual’s vulnerability to adhesion formation. The researchers collected genetic data from individuals with and without peritoneal adhesions, focusing on polymorphisms within genes involved in oxidative stress response pathways. Genetic variations, such as single-nucleotide polymorphisms (SNPs) within potential genes linked to oxidative stress, were analyzed using high-throughput genotyping techniques.

Statistical analyses were then performed to assess the association between specific genetic polymorphisms and adhesion susceptibility, controlling for potential confounding factors. The main objective of this study was to elucidate how genetic susceptibility to oxidative stress contributes to the pathogenesis of peritoneal adhesions. By identifying genetic variants associated with adhesion formation, the researchers aimed to uncover vital molecular mechanisms underlying oxidative stress-mediated adhesion formation and identify potential therapeutic targets for intervention. Understanding how genetic variations in oxidative stress response pathways influence adhesion susceptibility could provide insights into personalized risk assessment and treatment strategies for individuals predisposed to adhesions. Genetic susceptibility to oxidative stress represents an essential determinant of adhesion formation susceptibility. The etiology of peritoneal adhesions is dependent primarily on oxidative stress, and an individual’s susceptibility to adhesion development may be influenced by genetic polymorphisms within genes implicated in oxidative stress response pathways. Finding the genetic variants linked to adhesion susceptibility may help develop biomarkers for risk assessment and tailored adhesion prevention plans for those who are more likely to experience adhesion. Moreover, targeting oxidative stress response pathways may offer novel therapeutic approaches for preventing or treating peritoneal adhesions.

The Association of Epigenetic Changes with Adhesion Formation (#19, Table 5) study investigated the association between epigenetic changes and adhesion formation in peritoneal tissues. Epigenetic modifications are modifications to histone proteins and DNA that control the expression of genes without changing the underlying sequence of DNA. Histone modifications, non-coding RNAs, and DNA methylation are examples of epigenetic processes that are important in controlling cellular phenotypes and patterns of gene expression. The researchers collected peritoneal tissue samples from individuals with and without adhesion formation and analyzed epigenetic changes using various techniques. DNA methylation patterns were assessed using bisulfite sequencing or methylation-specific PCR assays. Histone modifications were examined using chromatin immunoprecipitation followed by sequencing (ChIP-seq) or histone modification-specific antibodies. Non-coding RNA expression levels were measured using quantitative RT-PCR or RNA sequencing. Bioinformatics analyses were then performed to identify differentially methylated regions, histone modification patterns, or dysregulated non-coding RNAs associated with adhesion formation [216,217].

This study’s primary goal was to clarify how epigenetic modifications contribute to the pathophysiology of peritoneal adhesions. The study’s objective was to understand better the molecular mechanisms underpinning adhesion formation and to find possible epigenetic biomarkers or therapeutic targets for intervention by detecting epigenetic changes linked to adhesion formation. Gaining knowledge about how gene expression patterns and cellular phenotypes in adhesion-prone tissues are impacted by epigenetic dysregulation may open our eyes to new perspectives on the pathophysiology of peritoneal adhesions. Essential factors of adhesion formation susceptibility are epigenetic modifications. 

The Functional Genomics Studies in Adhesion Formation (#20, Table 5) study aimed to investigate the functional genomics of adhesion formation, focusing on the comprehensive analysis of gene function and regulation in peritoneal tissues. Functional genomics systematically studies gene function, including gene expression, protein interactions, and regulatory mechanisms, to understand how genes contribute to biological processes. The main objective of this study was to elucidate the molecular mechanisms underlying adhesion formation through comprehensive functional genomics analyses. By integrating gene expression data with pathway analysis and functional validation experiments, the researchers aimed to identify critical genes and pathways involved in adhesion formation and gain insights into their roles in the pathogenesis of this condition. Understanding the functional genomics of adhesion formation could lead to identifying novel therapeutic targets and developing targeted interventions for preventing or treating peritoneal adhesions. This study represents a comprehensive investigation into the functional genomics of adhesion formation, providing valuable insights into the molecular mechanisms underlying this condition and potential therapeutic targets for intervention.

These investigations improve our knowledge of the intricate molecular processes underlying the creation of peritoneal adhesions. They draw attention to the interaction of genetic, epigenetic, and environmental variables in the pathophysiology of adhesions and pinpoint possible therapeutic targets and biomarkers for individualized risk assessment and intervention plans. More investigation is necessary to apply these discoveries to clinical settings and enhance results for individuals who are susceptible to peritoneal adhesions [218,219].

## 4. Bioinformatics and Systems Biology Applications

Bioinformatics and systems biology have revolutionized our ability to analyze, interpret, and integrate large-scale biological datasets, becoming indispensable tools in understanding complex conditions, such as peritoneal adhesion formation. By employing computational techniques and advanced algorithms, bioinformatics facilitates the systematic analysis of genomic, transcriptomic, proteomic, and metabolomic data, enabling researchers to identify critical genes, proteins, and regulatory pathways involved in adhesion pathology. Systems biology further enhances this capability by modeling the intricate networks and interactions among these molecular components, providing a comprehensive view of the biological processes driving adhesion formation. Leveraging tools, such as network analysis, pathway enrichment, predictive modeling, and machine learning algorithms, researchers can effectively uncover novel biomarkers, therapeutic targets, and personalized intervention strategies. The integration of bioinformatics and systems biology approaches, thus, represents a robust framework for transforming molecular-level discoveries into actionable clinical insights, ultimately improving prevention and management strategies for peritoneal adhesions. Additional in vivo research or perhaps clinical trials should be conducted expeditiously to validate the utilization of novel biological materials in addressing this substantial challenge. In the following years, biomarkers and biomarker screening can be used to detect patients predisposed to develop peritoneal adhesion after surgery. Those patients can presumably receive pre- or postintervention antiadhesion therapy. All these therapies are hopefully related to the rapid development of genetics and molecular medicine.

The developments in recent years regarding the peritoneal adhesions are listed in the table below (Table 6).

## 5. Therapeutic Strategies and Experimental Models

Despite significant advances in understanding the molecular mechanisms underlying peritoneal adhesion formation, effective clinical therapies remain limited. The need to translate molecular insights into therapeutic applications has prompted increasing research into gene- and pathway-specific interventions. A growing body of evidence supports the therapeutic potential of targeting key molecular pathways, such as VEGF, TGF-β, PDGF, COX-2, MMPs, and the fibrinolytic axis. However, the clinical translation of these targets has been uneven, with several experimental therapies still in the early investigational stages [234]. Clinical trials evaluating pharmacologic agents that inhibit angiogenesis and fibrotic signaling pathways are underway. For example, bevacizumab, a monoclonal antibody targeting VEGF-A, has shown promise in reducing peritoneal adhesion formation in gynecologic and gastrointestinal surgical contexts. Similarly, pirfenidone and fresolimumab, both inhibitors of TGF-β signaling, have demonstrated antifibrotic activity in other fibrotic diseases and are now being explored for adhesion prevention. Trials involving tissue plasminogen activator (tPA) or its analogs aim to enhance fibrinolysis and reduce fibrin-rich adhesion development. However, results remain inconsistent due to rapid clearance and systemic side effects [235].

Multiple gene and protein targets have been validated for adhesion modulation in animal models, particularly rodent and rabbit peritoneal injury models. The inhibition of PDGF receptors, VEGF receptors, or integrins, such as α5β1 using agents like ATN-161, has led to measurable reductions in adhesion severity. These findings reinforce the significance of these targets and support their progression into clinical trials. Murine models have also demonstrated that COX-2 selective inhibitors can reduce fibroblast activation and inflammatory signaling; although, careful balancing is necessary to avoid impairing routine wound healing [236].

RNA-based therapies represent a particularly promising avenue for the precision targeting of adhesion-promoting genes. Preclinical studies using small interfering RNAs (siRNAs) directed against TGF-β, COL1A1, PAI-1, and IL-6 have resulted in the significant downregulation of fibrotic gene expression, decreased collagen deposition, and improved tissue remodeling in peritoneal tissues. Additionally, antisense oligonucleotides (ASOs) have been employed to silence mRNAs associated with excessive matrix production and inflammation. These approaches offer a transient and reversible mechanism to inhibit pathogenic gene expression, making them attractive candidates for perioperative therapeutic delivery [237].

Perhaps the most transformative potential lies in the use of CRISPR/Cas9-based genome-editing technologies to modify or knock out adhesion-related genes directly. Experimental animal studies have demonstrated the successful editing of key genes, such as TGF-β, SMAD3, and MMP9, leading to reduced fibrosis and adhesion formation. These studies not only underscore the causal role of these genes but also open the door to potential long-term, possibly permanent, interventions. However, safety concerns, including off-target effects, immune responses, and the challenge of efficient in vivo delivery, remain critical barriers to clinical translation [238].

Moreover, the delivery systems used for these genetic and molecular therapies are evolving rapidly. Localized delivery through biodegradable hydrogels, liposomal nanoparticles, or thermosensitive polymer-based sprays enables the sustained and targeted exposure of therapeutic agents to injured peritoneal surfaces, minimizing systemic toxicity and enhancing therapeutic efficacy. These vehicles can be engineered to release gene silencers or inhibitors in response to inflammation or hypoxia, further enhancing specificity [239]. Future research should focus on integrating multi-omics profiling to stratify patients by genetic susceptibility and adhesion risk, enabling the personalization of antiadhesion therapies. Coupled with machine learning and bioinformatics, such approaches can refine target selection, predict therapeutic response, and optimize dosage regimens. The development of 3D tissue-engineered models and organoids, alongside advanced animal models, will further enable the testing of RNA- and CRISPR-based therapies under conditions that closely mimic human adhesion pathology [240].

In conclusion, the therapeutic landscape for peritoneal adhesions is rapidly expanding from general anti-inflammatory agents toward precise, gene-directed interventions. RNA interference, antisense oligonucleotides, and genome-editing technologies, like CRISPR/Cas9, offer unprecedented specificity in targeting the molecular drivers of adhesion formation. While these strategies remain in preclinical or early translational phases, they represent a promising frontier in the quest to reduce adhesion-related complications and improve postoperative outcomes.

## 6. Emerging Technologies and Future Directions

The landscape of peritoneal adhesion research is rapidly evolving with the integration of advanced technologies that extend beyond traditional molecular biology and histopathology. One of the most transformative developments is the application of multi-omics platforms—combining genomics, transcriptomics, proteomics, metabolomics, and epigenomics—to dissect the intricate molecular mechanisms driving adhesion formation. These high-resolution data layers offer a holistic view of cell behavior during injury, inflammation, and fibrosis, enabling researchers to identify key regulatory nodes that could serve as novel therapeutic targets [241].

Concurrently, artificial intelligence (AI) and machine learning (ML) are being increasingly harnessed to process the vast and complex datasets generated by multi-omics platforms—including genomics, transcriptomics, proteomics, and epigenomics. These computational tools can identify subtle patterns and interactions among genes, proteins, and signaling pathways that may not be apparent through conventional statistical analyses. In the context of peritoneal adhesions, AI and ML are being developed to predict individual susceptibility, identify molecular subtypes of fibrosis, and generate personalized risk profiles based on a patient’s genetic, surgical, and inflammatory background. One promising application is the use of supervised ML algorithms to integrate preoperative clinical data with molecular signatures, enabling the risk stratification of patients prior to abdominal or pelvic surgery. This could lead to personalized prophylactic interventions, such as targeted antifibrotic agents, tailored surgical techniques, or the selective use of adhesion barriers in high-risk cases. Additionally, unsupervised ML models can help cluster adhesion-related cases into biologically meaningful subgroups, which may reveal new pathophysiological mechanisms or therapeutic targets. Moreover, AI-driven predictive analytics can be applied intraoperatively and postoperatively through real-time decision support systems, incorporating live surgical data, inflammatory markers, or imaging parameters to adapt clinical strategies dynamically. Natural language processing (NLP) tools may even extract adhesion-related information from electronic health records (EHRs) to refine algorithms over time. Collectively, these innovations hold the potential to reduce the incidence, severity, and recurrence of adhesions, shifting the paradigm from reactive treatment to proactive, data-informed prevention [242].

On the experimental front, organ-on-chip systems and 3D bioprinted peritoneal tissue models are revolutionizing in vitro modeling of peritoneal adhesion formation. These platforms offer high-resolution microenvironments that mimic the biomechanical and biochemical properties of the human peritoneum. Organ-on-chip devices—microfluidic systems lined with human-derived mesothelial, fibroblast, and endothelial cells—can recapitulate shear stress, oxygen gradients, cytokine flux, and immune cell trafficking, providing dynamic insights into how mechanical forces and cellular interactions contribute to adhesion pathophysiology. These systems allow for precise manipulation of single variables, such as the application of TGF-β or hypoxia, to dissect pathway-specific responses and screen candidate therapeutics in a controlled, reproducible manner. Simultaneously, 3D bioprinting technologies have enabled the fabrication of peritoneal tissue constructs containing multiple layers of human cells embedded in hydrogels that mimic the native extracellular matrix (ECM). These constructs can be customized with pro-inflammatory cytokines, fibrin, or matrix metalloproteinase substrates to model the key features of postsurgical adhesion development. Importantly, they also offer the opportunity to test antiadhesive biomaterials, nanoparticles, or gene-editing interventions in a physiologically relevant human tissue context, circumventing the limitations of animal models in terms of species-specific immune responses and peritoneal architecture. In parallel, advanced imaging modalities are becoming integral to both experimental and clinical adhesion research. Techniques such as high-resolution magnetic resonance imaging (MRI) and molecular ultrasound now allow for the non-invasive, real-time visualization of adhesion formation, vascularization, and fibrotic remodeling. MRI with gadolinium contrast can delineate soft tissue planes and early fibrotic bands, while targeted ultrasound contrast agents can highlight areas of neovascularization and inflammation. These tools not only facilitate longitudinal monitoring of adhesion progression in vivo but also serve as endpoints in preclinical trials evaluating new therapeutics or materials [243,244].

Together, these emerging platforms offer a comprehensive and ethically sustainable framework for understanding, predicting, and preventing peritoneal adhesions. Their integration into the research pipeline promises to accelerate the translation of molecular insights into clinically applicable solutions, while reducing reliance on animal experimentation. The convergence of these technologies not only improves mechanistic understanding but also accelerates the translation of basic research into clinical solutions. Future efforts should focus on validating these tools in multicenter studies and integrating them into clinical trial pipelines. Embracing these innovations is essential to move from generalized antiadhesion measures toward precision medicine-based surgical care, ultimately improving long-term outcomes for patients undergoing abdominal and pelvic surgery [245,246,247,248].

## 7. Conclusions

Peritoneal adhesion formation remains a significant clinical challenge, contributing to postsurgical morbidity, infertility, and chronic pain. Increasing evidence demonstrates that adhesion development is a genetically determined, multifactorial process involving a wide range of molecules, such as growth factors (TGF-β, PDGF, VEGF), extracellular matrix components (fibronectin, integrins, cadherins), and inflammatory mediators (selectins, ICAM-1, VCAM-1). Advances in functional genomics, transcriptomics, and epigenetic profiling have provided a deeper understanding of the molecular and cellular mechanisms underlying peritoneal adhesion formation. Key processes include dysregulated wound healing, excessive extracellular matrix deposition, impaired fibrinolysis, and heightened inflammatory responses. Recent studies have identified novel genetic and epigenetic factors implicated in adhesion susceptibility, opening new avenues for biomarker discovery, personalized risk prediction, and targeted therapeutic interventions. Furthermore, regenerative medicine approaches, antifibrotic agents, and innovative surgical techniques offer promising strategies to reduce adhesion incidence. Nevertheless, significant gaps remain in translating these molecular insights into effective clinical therapies. Future research should prioritize comprehensive multi-omics studies to fully delineate adhesion pathogenesis, the development of precision medicine strategies tailored to individual genetic profiles, and clinical trials evaluating the efficacy of novel antiadhesion therapies.

Understanding the genetic determinism of peritoneal adhesions represents a critical step toward the development of predictive, preventive, and personalized surgical care. Continued interdisciplinary research is essential to mitigate the burden of adhesions and improve patient outcomes.

## Figures and Tables

**Figure 1 cimb-47-00475-f001:**
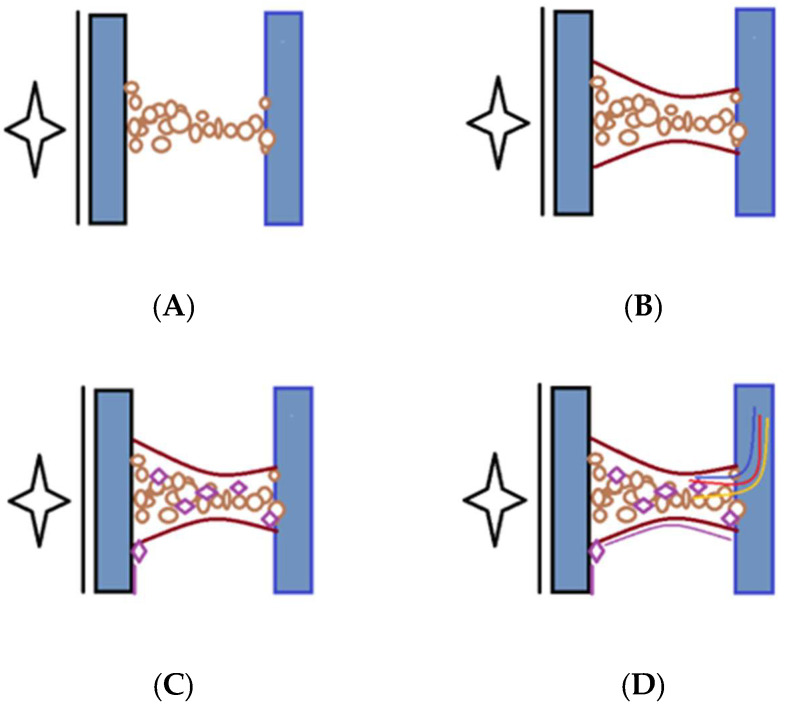
Peritoneal adhesion mechanism. Blue bands—mesothelium; circles—aggregated macrophages; brown lines—fibrin; star—myofibroblast to collagen fiber formation; and finally, mesothelium, red, blue, and yellow lines—artery, vein, and nerve, respectively. Adhesion development following surgery—preoperative overview of the peritoneal cavity. Major abdominal surgery is one example of a non-focal mesothelial injury that causes an uncontrollably large peritoneal macrophage aggregation (**A**) that acts as the nidus for the subsequent fibrin clot deposition (**B**). Fibrin deposition is promoted by coagulation and inflammation in concert. The intestine, for example, is currently affixed to the abdominal wall at anatomic (mesentery) and non-anatomic (adhesion) sites. Myofibroblasts, which migrate into the wound and fibrin clot to deposit extracellular matrix (ECM), such as collagen, are produced due to the mesenchymal to epithelial transition (**C**). The completion of adhesion development occurs when mesothelium covers the scar tissue. The lesion may become perfused and pain-sensitive due to blood vessels and nerves growing into it (**D**).

**Figure 2 cimb-47-00475-f002:**
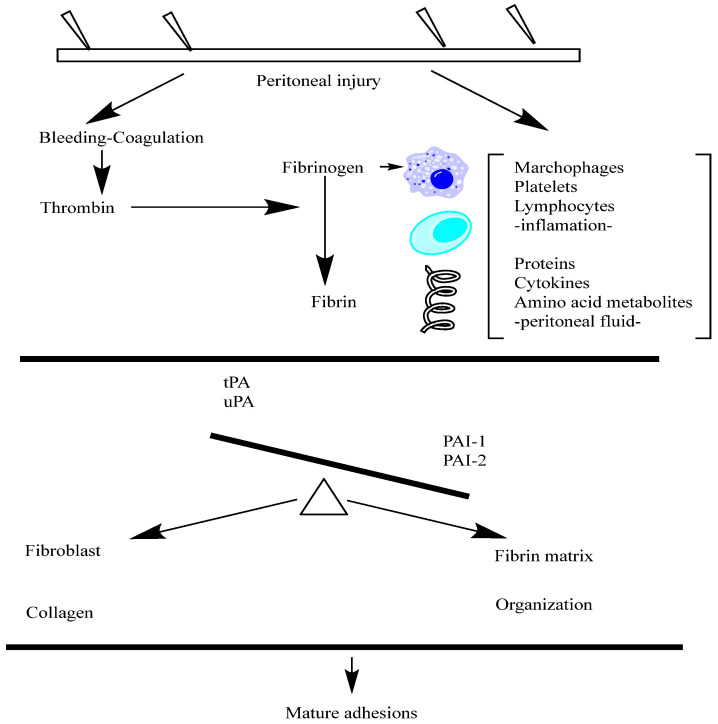
The peritoneal adhesion formation mechanism involves the fibrinogen pathway. tPA—tissue plasminogen activator, uPA—urokinase type plasminogen activator, PAI-1—plasminogen activator inhibitor 1, PAI-2—plasminogen activator inhibitor 2. Violet, blue, and black collared images: macrophage, lymphocyte, amino acid chains. When there is an imbalance between the plasminogen activator and plasminogen inhibitor factors, peritoneal adhesion is favored (as a result of collagen formation and organization).

**Table 1 cimb-47-00475-t001:** Proteins involved in peritoneal adhesion.

#	Protein	Description/Function	Pathway Involvement	Evidence Type	Ref.
1	Integrins	Transmembrane receptors facilitate cell–cell and cell–extracellular matrix interactions. They play a role in cell migration, proliferation, and tissue remodeling processes involved in peritoneal adhesion formation.	Integrin/FAK signaling	Preclinical (mouse, in vitro)	[22]
2	Cadherins (E-cadherin)	Calcium-dependent cell adhesion molecules that mediate homophilic interactions between cells. E-cadherin maintains tissue integrity and may contribute to peritoneal adhesion formation through its role in cell–cell adhesion.	Wnt/β-catenin, EMT	Preclinical (rat, human)	[23]
3	Selectins (P-selectin and E-selectin)	Cell adhesion molecules mediate initial interactions between leukocytes and endothelial cells during inflammation. Their involvement in peritoneal adhesion formation suggests a role in leukocyte recruitment and inflammatory processes at the peritoneal surface.	Inflammatory cascade	Preclinical (rat)	[23]
4	Immunoglobulin superfamily molecules (ICAM-1 and VCAM-1)	Cell adhesion molecules facilitate leukocyte trafficking and adhesion to endothelial cells. ICAM-1 and VCAM-1 may contribute to peritoneal adhesion formation by mediating leukocyte recruitment and interactions with mesothelial cells.	NF-κB-mediated inflammation	Preclinical (mouse, in vitro)	[24]
5	Hyaluronan (HA)	A significant glycosaminoglycan component of the extracellular matrix. HA can influence cell adhesion, migration, and tissue remodeling processes. Its involvement in peritoneal adhesion formation suggests a role in modulating cell–matrix interactions and tissue repair responses.	CD44/HA interaction	Preclinical (rat)	[25]
6	Fibronectin	An extracellular matrix glycoprotein is involved in cell adhesion, migration, and wound healing. Fibronectin may contribute to peritoneal adhesion formation by promoting cell–matrix interactions.	Integrin/ECM remodeling	Preclinical (mouse, cell lines)	[26]
7	Matrix metalloproteinases (MMPs)	Enzymes are involved in extracellular matrix degradation and remodeling. MMPs play a role in tissue repair and remodeling processes associated with peritoneal adhesion formation.	ECM remodeling pathway	Preclinical (human, rat)	[27]
8	Fibrinogen	A glycoprotein is involved in blood clot formation. Fibrinogen and its derivatives may contribute to peritoneal adhesion formation by promoting fibrin deposition.	Coagulation/fibrinolysis	Clinical and preclinical	[28]
9	Tissue plasminogen activator (tPA) and plasminogen activator inhibitor-1 (PAI-1)	Regulators of fibrinolysis. Alterations in tPA and PAI-1 levels may impact fibrin degradation and tissue remodeling processes associated with peritoneal adhesion formation.	Plasminogen activation system	Preclinical (mouse, human)	[29]
10	Transforming growth factor-beta (TGF-β)	A cytokine is involved in cell growth, differentiation, and extracellular matrix production. TGF-β signaling pathways may contribute to tissue fibrosis and peritoneal adhesion formation.	TGF-β/SMAD signaling	Preclinical and clinical	[30]
11	Vascular endothelial growth factor (VEGFA)	A signaling protein involved in angiogenesis, vasculogenesis, and vascular permeability. VEGF may contribute to peritoneal adhesion formation by promoting neovascularization and tissue remodeling.	VEGF/VEGFR pathway	Clinical and preclinical	[31]
12	Platelet-derived growth factor (PDGF)	A growth factor is involved in cell proliferation, migration, and angiogenesis. PDGF may contribute to peritoneal adhesion formation by stimulating fibroblast proliferation and extracellular matrix production.	PDGFR/PI3K-Akt	Preclinical (rabbit, human)	[32]
13	Interleukins (ILs)	Cytokines are involved in immune responses and inflammation. Certain interleukins, such as IL-1 and IL-6, may contribute to peritoneal adhesion formation by promoting inflammation and fibrosis.	JAK/STAT, NF-κB	Clinical and preclinical	[33]
14	Fibroblast growth factor (FGF1)	A family of growth factors involved in cell proliferation, migration, and differentiation. FGFs may contribute to peritoneal adhesion formation by stimulating fibroblast activity and extracellular matrix synthesis.	FGFR/MAPK pathway	Preclinical (mouse, in vitro)	[34]
15	Toll-like receptors (TLRs)	Pattern recognition receptors are involved in innate immune responses. TLRs may contribute to peritoneal adhesion formation by triggering inflammatory responses and modulating tissue repair processes.	TLR/MyD88/NF-κB	Preclinical (mouse, human)	[35]
16	Tenascin-C	An extracellular matrix glycoprotein is involved in cell adhesion, migration, and tissue remodeling. Tenascin-C expression may be upregulated during peritoneal injury and contribute to adhesion formation.	Wound healing, TGF-β signaling	Preclinical (mouse)	[36]
17	Platelet-derived growth factor (PDGF)	A mitogen is involved in cell proliferation and migration. PDGF signaling pathways may contribute to tissue repair and fibrosis associated with peritoneal adhesion formation.	VEGF signaling	Preclinical (rat)	[37]
18	Vascular endothelial growth factor (VEGFB)	A cytokine involved in angiogenesis and vascular permeability. VEGF may play a role in tissue repair and remodeling processes associated with peritoneal adhesion formation.	ECM remodeling balance	Preclinical (human, rat)	[38]
19	Tissue inhibitor of metalloproteinases (TIMPs)	Endogenous inhibitors of matrix metalloproteinases (MMPs). TIMPs regulate extracellular matrix turnover and tissue remodeling processes in peritoneal adhesion formation.	TGF-β/SMAD pathway	Preclinical (mouse)	[39]
20	Fibroblast growth factor (FGF2)	A family of growth factors involved in cell proliferation, migration, and tissue repair. FGF signaling pathways may contribute to fibrosis and adhesion formation in the peritoneum.	ECM synthesis/fibrosis	Preclinical (human, animal)	[40]

**Table 2 cimb-47-00475-t002:** Summary of genomic and molecular insights.

Gene/Protein	Pathway/Function	Experimental Model	Therapeutic Relevance
TGF-β (TGF-β/β2/β3)	Fibrosis, ECM deposition	Mouse, human fibroblasts	TGF-β inhibitors (therapeutic modulation)
COX-2	Inflammation, fibroblast activation	Rat models, cell cultures	COX-2 selective inhibitors (anti-inflammatory therapy)
tPA/PAI-1	Fibrinolysis balance	Human peritoneal tissues, rat models	Fibrinolytic agents
VEGF	Angiogenesis, vascular remodeling	Rat/mouse adhesion models	Anti-VEGF therapy (e.g., bevacizumab)
PDGF	Fibroblast proliferation	Rabbit adhesion models, human tissues	PDGF receptor inhibitors
MMPs/TIMPs	ECM remodeling	Mouse, human cell models	Selective MMP inhibitors

Abbreviations: ECM (extracellular matrix), MMP (matrix metalloproteinases), TIMPs (tissue inhibitors of metalloproteinases), TGF-β (transforming growth factor-beta), COX-2 (cyclooxygenase-2), VEGF (vascular endothelial growth factor), PDGF (platelet-derived growth factor), tPA (tissue plasminogen activator), PAI-1 (plasminogen activator inhibitor-1) [45,46,47,48,49,50].

**Table 3 cimb-47-00475-t003:** Genes implicated in the peritoneal adhesion process.

#	Gene	Related Protein	Function/Role	Pathway Involvement	Evidence Type	Ref
1	Transforming Growth Factor-beta (TGF-β)	TGF-β, TGF-β2, TGF-β3	Fibrosis, fibroblast activation	TGF-β/SMAD	Clinical and Preclinical	[126]
2	Vascular Endothelial Growth Factor (VEGF)	VEGF-A, VEGF-B, VEGF-C, VEGF-D, placental growth factor (PlGF)	Angiogenesis, vascular permeability	VEGF/VEGFR	Clinical and Preclinical	[127]
3	Platelet-Derived Growth Factor (PDGF)	PDGF-A, PDGF-B, PDGF-C, PDGF-D	Fibroblast proliferation, ECM production	PDGF/PI3K-Akt	Preclinical	[128]
4	Fibroblast Growth Factor (FGF)	FGF1 (acidic FGF), FGF2 (basic FGF), FGF3, FGF4, FGF5	Cell proliferation, angiogenesis	FGFR/MAPK	Preclinical	[88]
5	Tissue Plasminogen Activator (tPA)	Tissue plasminogen activator (tPA)	Fibrinolysis activation	Plasminogen system	Preclinical	[129]
6	Plasminogen Activator Inhibitor-1 (PAI-1)	Plasminogen activator inhibitor-1 (PAI-1)	Inhibits fibrinolysis	Plasminogen system	Preclinical	[130]
7	Integrins	Various alpha and beta subunits forming different integrin receptors	Cell adhesion, migration	Integrin/FAK	Preclinical	[131]
8	Cadherins	E-cadherin (CDH1), N-cadherin (CDH2), P-cadherin (CDH3)	Cell–cell adhesion, EMT regulation	Wnt/β-catenin, EMT	Preclinical	[128]
9	Selectins	P-selectin, E-selectin	Leukocyte–endothelial interaction	Inflammation	Preclinical	[132]
10	Immunoglobulin Superfamily Molecules	Intercellular adhesion molecule 1 (ICAM-1), vascular cell adhesion molecule 1 (VCAM-1)	Leukocyte adhesion, inflammation	NF-κB, cytokine response	Preclinical	[89]
11	Hyaluronan (HA)	Hyaluronan synthases (HAS), hyaluronidases (HYAL)	ECM hydration, structural matrix remodeling	CD44/HA interaction	Preclinical	[133]
12	Fibronectin	Fibronectin	ECM scaffold, fibroblast migration	ECM-integrin	Preclinical	[132]
13	Matrix Metalloproteinases (MMPs)	MMP-1, MMP-2, MMP-3, MMP-9, MMP-13,	ECM degradation	ECM remodeling	Preclinical	[134]
14	Tissue Inhibitors of Metalloproteinases (TIMPs)	TIMP-1, TIMP-2, TIMP-3, TIMP-4	Inhibit MMPs, control ECM turnover	MMP/TIMP balance	Preclinical	[42]
15	Toll-Like Receptors (TLRs)	Various Toll-like receptors (TLR1, TLR2, TLR4,	Innate immune activation	TLR/MyD88/NF-κB	Preclinical	[94]
16	Tenascin-C	Tenascin-C	Fibroblast migration, ECM modulation	Wound healing, inflammation	Preclinical	[135]

**Table 4 cimb-47-00475-t004:** Areas of advancement in peritoneal adhesion research.

#	Development	Description	Ref
1	Genetic Studies	Continued research into the genetic determinants of peritoneal adhesions, including identification of novel genetic risk factors and pathways involved in adhesion formation.	[166]
2	Biomarkers	The discovery and validation of biomarkers associated with peritoneal adhesions for early diagnosis, prognosis, and adhesion development and recurrence monitoring.	[167]
3	Therapeutic Targets	The identification of novel molecular targets and signaling pathways implicated in peritoneal adhesions has resulted in the development of targeted medicines and interventions for adhesion prevention and therapy.	[168]
4	Regenerative Medicine	Advancements in regenerative medicine approaches, including stem cell therapy, tissue engineering, and biocompatible scaffolds, are being made to promote tissue repair and reduce adhesion formation following surgery or injury.	[169]
5	Minimally Invasive Techniques	Innovation in surgical techniques and devices aimed at minimizing tissue trauma, inflammation, and adhesion formation during abdominal and pelvic surgeries, such as laparoscopy and robotic-assisted surgery.	[170]
6	Drug Delivery Systems	The development of localized drug delivery systems and formulations, including hydrogels, nanoparticles, and coatings, for delivering antiadhesive agents or therapeutic drugs directly to the site of adhesion formation.	[171]
7	Imaging Modalities	Technological advancements, such as magnetic resonance imaging (MRI) and ultrasound, have made improvements in the visualization and characterization of peritoneal adhesions possible, enabling early diagnosis and precise adhesion severity evaluation.	[172]
8	Clinical Guidelines	Established evidence-based clinical guidelines and protocols for managing peritoneal adhesions, including standardized approaches to adhesion prevention, surgical techniques, and postoperative care.	[173]
9	Patient Education and Awareness	Increased awareness and education initiatives aimed at patients, healthcare providers, and policymakers regarding the risks, complications, and preventive measures associated with peritoneal adhesions.	[174]
10	Multidisciplinary Collaboration	The promotion of interdisciplinary collaboration among surgeons, researchers, engineers, and industry partners to address the complex challenges of peritoneal adhesions through integrated research, innovation, and clinical practice.	[175]

**Table 5 cimb-47-00475-t005:** Further research areas in peritoneal adhesion.

#	Study	Ref
1	Genetic Polymorphisms Associated with Adhesion Formation	[176]
2	Gene Expression Profiling in Adhesion Formation	[177]
3	Association Studies of Candidate Genes	[178]
4	Genome-wide Association Studies (GWAS) in Peritoneal Adhesions	[179]
5	Functional Genomics Studies	[180]
6	Epigenetic Regulation of Genes Involved in Adhesion Formation	[181]
7	Association of Inflammatory Genes with Adhesion Formation	[182]
8	The Function of Growth Factors and Their Receptors in the Formation of Adhesion	[183]
9	Genetic Variation in Extracellular Matrix Proteins and Adhesion Formation	[184]
10	Epigenetic Modifications and Adhesion Formation	[185]
11	Impact of miRNA Dysregulation on Adhesion Formation	[186]
12	Role of Cytokine Signaling Pathways in Adhesion Formation	[187]
13	Identification of Novel Genetic Risk Loci for Adhesion Formation	[188]
14	MicroRNA (miRNA) Profiling in Adhesion Formation	[189]
15	Genetic Variation in Cell Adhesion Molecules and Adhesion Formation	[190]
16	Gene Expression Profiling in Peritoneal Tissue	[191]
17	Impact of Genetic Polymorphisms on Inflammatory Signaling Pathways	[192]
18	Genetic Susceptibility to Oxidative Stress and Adhesion Formation	[193]
19	Association of Epigenetic Changes with Adhesion Formation	[194]
20	Functional Genomics Studies in Adhesion Formation	[195]

**Table 6 cimb-47-00475-t006:** The latest development in experimental research regarding peritoneal adhesion.

#	Development	Description	Ref
1	Identification of Genetic Risk Factors	Researchers are conducting genome-wide association studies (GWAS) and candidate gene studies to identify genetic variations associated with peritoneal adhesions.	[220]
2	Gene Expression Studies	Transcribing tissues from people with peritoneal adhesions using transcriptome analysis to find dysregulated genes and pathways.	[221]
3	Epigenetic Mechanisms	Investigation of DNA methylation patterns and histone modifications in individuals with peritoneal adhesions.	[222]
4	Animal Models	Animal models, such as rodents and pigs, are utilized to study the genetic basis of peritoneal adhesions.	[223]
5	Translational Research	Translation of genetic findings into clinical applications, including diagnostic tests and targeted therapies.	[224]
6	Functional Genomics Studies	Investigate the functional role of identified genetic variants using cell culture models and gene editing techniques.	[225]
7	Multi-Omics Approaches	Integration of data from proteomics, metabolomics, transcriptomics, and genomes to offer a thorough knowledge of the molecular processes behind peritoneal adhesions.	[226]
8	Machine Learning and Bioinformatics	Machine learning algorithms and bioinformatics tools are used to analyze large-scale genetic and clinical datasets and identify predictive biomarkers and therapeutic targets.	[227]
9	Genetic Epidemiology Studies	Population-based studies to investigate the prevalence of genetic risk factors for peritoneal adhesions across different populations and ethnicities.	[228]
10	Pharmacogenomics and Personalized Medicine	Exploration of genetic factors influencing individual responses to pharmacological interventions for preventing or treating peritoneal adhesions.	[135]
11	Longitudinal Genetic Studies	Long-term studies track genetic variations and their impact on peritoneal adhesion formation and recurrence.	[229]
12	Functional Validation of Candidate Genes	Functional validation of candidate genes associated with peritoneal adhesions using in vitro and in vivo models.	[230]
13	Identification of Genetic Biomarkers	Discovery of genetic biomarkers for early detection, prognosis, and monitoring of peritoneal adhesions.	[231]
14	Gene-Environment Interactions	They were investigating how genetic predisposition and environmental variables interact to generate peritoneal adhesions.	[232]
15	Network Analysis of Genetic Pathways	Network analysis to elucidate interactions among genes and pathways implicated in peritoneal adhesion formation.	[233]

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
