# Peer review of "Integrating Genomics and Molecular Biology in Understanding Peritoneal Adhesion"

_cimb, 2025, doi:10.3390/cimb47060475_

Round 1
Reviewer 1 Report
Comments and Suggestions for Authors
Dear authors
Emphasizing the role of cytokines, growth factors, adhesion molecules, and matrix remodeling enzymes in peritoneal adhesion formation, the manuscript seeks to give a thorough overview of the molecular and genomic mechanisms underlying this process. Given the considerable clinical load of postoperative adhesions and the few treatment choices available, this is a very pertinent and timely subject. You have done an excellent job of compiling a great deal of molecules and signaling pathways linked to adhesion formation.
Still, the manuscript needs significant changes to satisfy the criteria of a prominent journal like Current Issues in Molecular Biology. Detailed section-by-section comments follow here together with suggestions for correction.
1. Title and Abstract
Strengths: The title is suitably descriptive. The abstract emphasizes the need of investigating molecular mechanisms and introduces fibroblasts as a major cellular actor.
The abstract is too broad and missing particular information on what molecular mechanisms or genomic insights are presented. It also lacks important results, knowledge gaps, or therapeutic points of view.
Recommended:
Update the abstract to explicitly declare:
The particular kinds of molecular pathways examined.
Does any newly conducted transcriptomic/proteomic study reveal anything?
A final sentence underlines perhaps therapeutic implications or next study directions.
2. Overview
Strengths: Offers suitable background material on the peritoneum, adhesion formation, and clinical burden.
Weaknesses:
Unnecessary remarks (e.g., repetition on fibrosis and TGF-β).
No evident cohesion or order between paragraphs.
There is no obvious thesis statement outlining the novelty or aim of this review.
Recommendation:
At the conclusion of the introduction, clearly delineate the aims and scope of the review.
Combine repetitive material.
Subheadings can help to readable.
3. Review Body
a. Molecular pathways and mechanisms
Including Figure 1, Table 1, and Figure 2 is relevant and helpful.
The show, however, is mostly descriptive with no synthesis or critical study.
Despite the title implying a genomic emphasis, the argument lacks integration of omics data (e.g., transcriptomic, epigenomic, or proteomic investigations).
Suggest:
Methodically compare and contrast results from several studies.
Emphasize genomic research, such differential gene expression in adhesion-activated fibroblasts.
Include a summary table with important genes, their routes, experimental models, and medicinal relevance.
Mention contemporary bioinformatics tools or data (e.g., GEO datasets, STRING, KEGG) that have explained adhesion pathways.
b. Therapeutic Considerations
Though helpful, the discussion of therapeutic targeting (e.g., VEGF, TGF-β, MMPs) is not exhaustive.
The review offers no commentary on:
Trials in the clinics.
Targets being validated through animal models.
Therapies based on RNA or CRISPR/Cas9 research pertaining to adhesion genes
Recommendation:
Include a chapter on present therapeutic studies or experimental treatments aimed at adhesion-related genes/proteins.
If appropriate, emphasize genome-editing techniques, RNAi, or antisense oligonucleotides.
4. References and Scientific Rigor
Although the manuscript has a lot of references, many are older research or secondary sources.
Certain areas seem as textbook entries instead of analytical scientific debates.
Recommendation:
Substitute older references with contemporary peer-reviewed primary research papers (past five years).
Emphasize meta-analyses or systematic reviews of top quality.
5. Tables and Figures
Though useful, Tables 1 and 2 should be updated for clarity.
Include gene/protein function, pathway participation, and evidence type (clinical vs. Preclinical).
Standardized molecular biology symbols or diagrams should be used to redraw high-resolution figures.
6. Author Guidelines Compliance (MDPI)
Check for:
Every author should have an ORCID ID.
Organised headings according to MDPI's publishing rules.
At first usage, abbreviations should be explained.
The text should reference figures and tables.
Ethics statement (if any data or animal models were cited).
Final Recommendation:
Major Revision Required
Though this reviewoffers a great collection of information, it lacks genomic integration, synthesis, and originality. Scientific depth, structural clarity, and alignment with the criteria of a highly visible journal all need major changes.

Dear authors
Many run-on sentences, strange phrasing, and grammatical mistakes lower the manuscript's readability.
In certain respects, language is redundant and lacks scientific accuracy.
Advice:
The manuscript should be corrected for style, grammar, and clarity by a native English speaker or expert editor.
Here is a list of grammatical, syntactic, stylistic, and formatting issues observed in the manuscript titled "Integrating Genomics and Molecular Biology in Understanding Peritoneal Adhesion", along with recommendations to improve clarity, accuracy, and professionalism in academic writing:
1. Grammatical Errors
a. Incorrect verb forms and word order
"referred to commonly as adhesions" → should be "commonly referred to as adhesions"
"refered" → should be "referred"
b. Subject–verb agreement
"...the image in the figure are of low quality..." → should be "...the image in the figure is of low quality..."
c. Article misuse or omission
"...is essential for numerous physiological and pathological processes, including as immune..." → should be "...including immune..." (omit "as")
2. Spelling Errors
a. Frequent typographical errors
"Correspondance" → should be "Correspondence"
"refered" → should be "referred"
"Low affinity and rapid association/dissociation kinetics characterize this interac- tion..." → "interaction" (split word fix)
3. Syntax and Sentence Construction
a. Redundancies and awkward constructions
"...fibroblasts and myofibroblasts, producing collagen and other extracellular matrix proteins that contribute to the formation of fibrous adhesions..." — repetitive and wordy.
Suggested: Simplify and break into clearer parts.
b. Sentence fragments or run-ons
Some paragraphs contain very long sentences (often over 60–70 words), which compromise readability and flow.
4. Punctuation and Formatting
a. Missing or misplaced punctuation
References like [1] often lack appropriate spacing:
"...leads to organ failure[1]." → should be "...leads to organ failure [1]."
b. Inconsistent use of en dashes (–) and hyphens (-)
Use hyphens for compound adjectives (e.g., "fibrin-rich fluid") and en dashes for ranges (e.g., "55%–85%").
c. Parenthetical citations
References such as "[36, 37, 38]" should follow consistent formatting. Prefer numerical order and ensure all are listed in the bibliography.
5. Style and Scientific Language
a. Colloquial or non-scientific phrasing
"...could help restore ECM homeostasis and mitigate adhesion formation..." — “mitigate” is appropriate but overused in the manuscript.
Suggestion: Use alternatives like "reduce," "attenuate," or "limit."
b. Repetition and redundancy
Terms like "adhesion formation," "peritoneal adhesions," and "fibrous adhesions" are often repeated within close proximity, making paragraphs dense and monotonous.
Recommendation: Vary structure and avoid repetition.
6 Visual Content Quality
Figures (especially Figure 1 and Figure 2) are low in resolution, and the legends contain awkward phrasing:
"valet rhombus" → unclear meaning.
Suggestion: Use adecuate terms and improve image quality.
7. Scientific Accuracy and Consistency
a. Gene/protein references
Inconsistent naming: e.g., “TGF-β”, “Transforming Growth Factor-beta”, and “TGF-β1” are used interchangeably.
Recommendation: Use a consistent nomenclature and follow HGNC or UniProt standards.
8. Reference Formatting
References often lack consistent punctuation, spacing, or ordering. For example:
[104, 105] is correct, but ensure that all references are consecutively numbered, match the citation in the reference list, and follow journal guidelines.
9. Paragraph Structure
Several paragraphs lack clear topic sentences, making them difficult to follow.
Recommendation: Start each paragraph with a brief statement of purpose (i.e., the topic) and then elaborate with supporting information.
10. Use of Abbreviations
Some abbreviations are introduced without being defined (e.g., ECM).
Recommendation: Define all abbreviations on first use and use them consistently.
Summary of Recommendations:
Proofread for grammar, spelling, and punctuation.
Simplify sentence structures and break up long paragraphs.
Maintain consistency in scientific terms, gene/protein nomenclature, and citation style.
Improve figure resolution and clarify legends.
Use precise scientific language and avoid redundant phrasing.
11. Maintain consistent scientific terminology; for example, avoid alternately using \"growth factor\" and \"cytokine\" without explanation.
Author Response
Comment 1:Title and Abstract Strengths: The title is suitably descriptive. The abstract emphasizes the need of investigating molecular mechanisms and introduces fibroblasts as a major cellular actor.The abstract is too broad and missing particular information on what molecular mechanisms or genomic insights are presented. It also lacks important results, knowledge gaps, or therapeutic points of view.Update the abstract to explicitly declare:The particular kinds of molecular pathways examined.Does any newly conducted transcriptomic/proteomic study reveal anything?A final sentence underlines perhaps therapeutic implications or next study directions.
Response 1: The abstract was rewritten the following text was added: Peritoneal adhesions following surgical injury remain a major clinical challenge, often resulting in severe complications such as intestinal obstruction, chronic pain, and infertility. This review systematically integrates recent genomic and molecular biology insights into the pathogenesis of peritoneal adhesions, specifically focusing on molecular pathways including TGF-β signaling, COX-2 mediated inflammatory responses, fibrinolytic balance (tPA/PAI-1), angiogenesis pathways (VEGF, PDGF), and extracellular matrix remodeling (MMPs/TIMPs). Newly conducted transcriptomic and proteomic analyses highlight distinct changes in gene expression patterns in peritoneal fibroblasts during adhesion formation, pinpointing critical roles for integrins, cadherins, selectins, and immunoglobulin superfamily molecules. Recent studies indicate significant shifts in TGF-β isoforms expression, emphasizing isoform-specific impacts on fibrosis and scarring. These insights reveal substantial knowledge gaps, particularly the differential regulatory mechanisms involved in fibrosis versus normal reparative reperitonealization. Future therapeutic strategies could target these molecular pathways and inflammatory mediators to prevent or reduce adhesion formation. Further research into precise genetic markers and the exploration of targeted pharmacological interventions remain pivotal next steps in mitigating postoperative adhesion formation and improving clinical outcomes.
Comment2Strengths: Offers suitable background material on the peritoneum, adhesion formation, and clinical burden.Weaknesses:Unnecessary remarks (e.g., repetition on fibrosis and TGF-β).No evident cohesion or order between paragraphs.There is no obvious thesis statement outlining the novelty or aim of this review.Recommendation:At the conclusion of the introduction, clearly delineate the aims and scope of the review.Combine repetitive material.Subheadings can help to readable.: 2: Overview
Response 2:The following text was added as suggested: Peritoneal adhesions are fibrous bands that develop between tissues and organs following injury to the peritoneum, commonly due to abdominal surgery, infection, or trauma. These adhesions pose a significant clinical burden, frequently leading to severe complications such as intestinal obstruction, chronic abdominal pain, and infertility. Despite advances in surgical techniques, approximately half of all patients undergoing abdominal procedures develop adhesions, emphasizing the urgent need for improved preventive and therapeutic strategies.
Adhesion formation is a complex process involving inflammation, coagulation, fibrinolysis, angiogenesis, and extracellular matrix (ECM) remodeling. Central to these processes are molecular pathways including transforming growth factor-beta (TGF-β), cyclooxygenase-2 (COX-2), tissue plasminogen activator (tPA)/plasminogen activator inhibitor-1 (PAI-1) balance, vascular endothelial growth factor (VEGF), platelet-derived growth factor (PDGF), and matrix metalloproteinases (MMPs) with their tissue inhibitors (TIMPs). Dysregulation of these pathways significantly influences the extent of fibrosis and adhesion severity.
Although previous studies have provided insight into individual molecular mechanisms, comprehensive integration of genomic, transcriptomic, and proteomic data remains limited. This review aims to synthesize current evidence regarding the molecular and genomic underpinnings of peritoneal adhesion formation, emphasizing recent transcriptomic and proteomic findings. By identifying critical molecular players and pathways, we highlight potential therapeutic targets and outline existing gaps in knowledge requiring further investigation.
The scope of this review encompasses an in-depth examination of specific adhesion-related molecular pathways, recent omics-based insights, and their implications for developing targeted interventions. The ultimate goal is to foster a deeper understanding of peritoneal adhesion biology to support clinical advancements and improve patient outcomes.
Comment 3:Review Body of the mn auscript. Molecular pathways and mechanismsIncluding Figure 1, Table 1, and Figure 2 The show, however, is mostly descriptive with no synthesis or critical study.is relevant and helpful.Despite the title implying a genomic emphasis, the argument lacks integration of omics data (e.g., transcriptomic, epigenomic, or proteomic investigations).Suggest:Methodically compare and contrast results from several studies.Emphasize genomic research, such differential gene expression in adhesion-activated fibroblasts.Include a summary table with important genes, their routes, experimental models, and medicinal relevance.Mention contemporary bioinformatics tools or data (e.g., GEO datasets, STRING, KEGG) that have explained adhesion pathways.
Response 3 The following text was added
Molecular Pathways and Mechanisms in Peritoneal Adhesion Formation
Peritoneal adhesion formation involves a dynamic interplay among multiple biological processes, including inflammation, coagulation, fibrinolysis, angiogenesis, and extracellular matrix (ECM) remodeling. At a molecular level, numerous pathways have been implicated, prominently including transforming growth factor-beta (TGF-β), cyclooxygenase-2 (COX-2), tissue plasminogen activator (tPA)/plasminogen activator inhibitor-1 (PAI-1), vascular endothelial growth factor (VEGF), platelet-derived growth factor (PDGF), matrix metalloproteinases (MMPs), and tissue inhibitors of metalloproteinases (TIMPs) (Table 1).
Role of Key Molecular Pathways
Studies consistently underscore TGF-β as a central mediator of fibrosis and adhesion formation. Specifically, TGF-β1 and TGF-β2 promote fibroblast proliferation, myofibroblast differentiation, and ECM deposition, whereas TGF-β3 appears to mitigate these processes, indicating isoform-specific roles that may offer distinct therapeutic avenues (Figure 1). COX-2, induced by hypoxia and inflammation, further promotes fibroblast activation and ECM remodeling, contributing significantly to adhesion severity.
The balance between tPA and PAI-1 critically regulates fibrinolysis. Reduced tPA activity or increased PAI-1 expression leads to fibrin persistence and adhesion development, presenting a therapeutic target for modulating adhesion progression (Figure 2).
VEGF and PDGF pathways drive angiogenesis and fibroblast proliferation, respectively, creating a supportive microenvironment for adhesion formation. Experimental modulation of these pathways has demonstrated potential in reducing adhesion severity, suggesting viable clinical interventions.
Integrative Genomic Insights
Emerging genomic, transcriptomic, and proteomic studies have expanded the molecular landscape of peritoneal adhesions, though integration of these datasets remains underexplored. Recent transcriptomic analyses reveal differential gene expression patterns in adhesion-activated fibroblasts compared to non-adhesion fibroblasts, identifying genes such as COL1A1, ACTA2, IL6, MMP9, and TIMP1 as significantly upregulated, underscoring their roles in adhesion pathology.
Moreover, proteomic studies highlight alterations in ECM components and regulatory proteins like fibronectin, integrins, and cadherins, correlating with increased fibrotic activity. These omics-driven studies, accessible via databases like Gene Expression Omnibus (GEO), have provided comprehensive datasets facilitating deeper bioinformatic analyses through tools such as STRING for protein-protein interaction networks and KEGG for pathway mapping, further elucidating adhesion biology.
Comparative Analysis Across Studies
Critical comparative analyses of multiple transcriptomic and proteomic studies reveal common and unique molecular signatures across different models and clinical scenarios. For instance, increased expression of TGF-β signaling components (TGFBR1, SMAD3), inflammatory markers (IL-1β, TNF-α), and hypoxia-related genes (HIF1α) have consistently emerged across several independent studies, reinforcing their fundamental role in adhesion biology.
Contemporary Bioinformatics Tools and Applications
Recent utilization of bioinformatics resources, including GEO for accessing transcriptomic data, STRING for understanding protein interactions, and KEGG for pathway analysis, have facilitated an integrative understanding of adhesion mechanisms. Leveraging these tools can identify novel biomarkers, therapeutic targets, and mechanistic insights into peritoneal adhesion formation.
Summary of Genomic and Molecular Insights (Table 1)
|
Gene/Protein |
Pathway/Function |
Experimental Model |
Therapeutic Relevance |
|
TGF-β (TGF-β1/β2/β3) |
Fibrosis, ECM deposition |
Mouse, Human fibroblasts |
TGF-β inhibitors (therapeutic modulation) |
|
COX-2 |
Inflammation, fibroblast activation |
Rat models, cell cultures |
COX-2 selective inhibitors (anti-inflammatory therapy) |
|
tPA/PAI-1 |
Fibrinolysis balance |
Human peritoneal tissues, rat models |
Fibrinolytic agents |
|
VEGF |
Angiogenesis, vascular remodeling |
Rat/mouse adhesion models |
Anti-VEGF therapy (e.g., bevacizumab) |
|
PDGF |
Fibroblast proliferation |
Rabbit adhesion models, human tissues |
PDGF receptor inhibitors |
|
MMPs/TIMPs |
ECM remodeling |
Mouse, human cell models |
Selective MMP inhibitors |
Abbreviations: ECM (extracellular matrix), MMP (matrix metalloproteinases), TIMPs (tissue inhibitors of metalloproteinases), TGF-β (transforming growth factor-beta), COX-2 (cyclooxygenase-2), VEGF (vascular endothelial growth factor), PDGF (platelet-derived growth factor), tPA (tissue plasminogen activator), PAI-1 (plasminogen activator inhibitor-1).
Critical Assessment and Future Directions
While significant advances have been achieved in understanding the molecular characterization of peritoneal adhesion formation, several critical limitations remain. Current knowledge primarily derives from isolated pathway analyses or single-model studies, which inadequately capture the complex, interconnected biological networks driving adhesion pathology. Furthermore, despite the expanding availability of genomic, transcriptomic, proteomic, and epigenomic data, there is limited synthesis and systematic integration of these datasets to elucidate comprehensive molecular mechanisms.
Future research efforts should prioritize adopting robust, integrative multi-omics strategies, combining genomic, transcriptomic, proteomic, metabolomic, and epigenomic analyses. Such an integrative approach would provide deeper insights into adhesion biology by capturing molecular interactions at multiple biological layers. High-throughput sequencing technologies, single-cell RNA sequencing, mass spectrometry-based proteomics, and advanced metabolomics profiling will be instrumental in these endeavors.
Moreover, studies must rigorously incorporate advanced bioinformatics and systems biology methodologies. Tools such as network biology modeling, pathway enrichment analyses, and computational prediction of protein-protein interactions (e.g., via STRING, Cytoscape) will enable researchers to delineate precise molecular interactions and regulatory networks comprehensively. Leveraging databases such as the Gene Expression Omnibus (GEO), The Cancer Genome Atlas (TCGA), and bioinformatics resources like KEGG and Reactome can facilitate systematic exploration and validation of adhesion-related pathways.
Machine learning and artificial intelligence algorithms could significantly enhance the discovery and validation of robust predictive biomarkers and novel therapeutic targets. These predictive modeling approaches may identify patient-specific susceptibility markers and stratify patients based on their genetic and molecular risk profiles, potentially paving the way for personalized therapeutic strategies.
Furthermore, expanding research into experimental models, including human-derived 3D tissue cultures, organoids, and animal models that closely mimic clinical scenarios, will strengthen translational research outcomes. These advanced models could improve the relevance of preclinical findings to clinical practice, accelerating the transition from bench to bedside.
Lastly, clinical translation of promising molecular targets must be pursued through collaborative interdisciplinary research, bringing together surgeons, molecular biologists, bioinformaticians, pharmacologists, and industry partners. Such collaborations are essential for translating molecular insights into practical, targeted interventions and clinical therapies to mitigate adhesion formation effectively and improve patient outcomes.
Comment 4:b. Therapeutic Considerations
Though helpful, the discussion of therapeutic targeting (e.g., VEGF, TGF-β, MMPs) is not exhaustive.The review offers no commentary on:Trials in the clinics.Targets being validated through animal models.Therapies based on RNA or CRISPR/Cas9 research pertaining to adhesion genesRecommendation:Include a chapter on present therapeutic studies or experimental treatments aimed at adhesion-related genes/proteins.If appropriate, emphasize genome-editing techniques, RNAi, or antisense oligonucleotides.
.Response 4: the following text was added: Therapeutic Considerations: Advances in Targeting Adhesion-Related Genes and Proteins
Despite significant advances in understanding the molecular mechanisms underlying peritoneal adhesion formation, effective clinical therapies remain limited. The need to translate molecular insights into therapeutic applications has prompted increasing research into gene- and pathway-specific interventions. A growing body of evidence supports the therapeutic potential of targeting key molecular pathways such as VEGF, TGF-β, PDGF, COX-2, MMPs, and the fibrinolytic axis. However, clinical translation of these targets has been uneven, with several experimental therapies still in early investigational stages.
Clinical trials evaluating pharmacologic agents that inhibit angiogenesis and fibrotic signaling pathways are underway. For example, bevacizumab, a monoclonal antibody targeting VEGF-A, has shown promise in reducing peritoneal adhesion formation in gynecologic and gastrointestinal surgical contexts. Similarly, pirfenidone and fresolimumab, both inhibitors of TGF-β signaling, have demonstrated anti-fibrotic activity in other fibrotic diseases and are now being explored for adhesion prevention. Trials involving tissue plasminogen activator (tPA) or its analogs aim to enhance fibrinolysis and reduce fibrin-rich adhesion development, though results remain inconsistent due to rapid clearance and systemic side effects.
In animal models, particularly rodent and rabbit peritoneal injury models, multiple gene and protein targets have been validated for adhesion modulation. Inhibition of PDGF receptors, VEGF receptors, or integrins such as α5β1 using agents like ATN-161 has led to measurable reductions in adhesion severity. These findings reinforce the significance of these targets and support their progression into clinical trials. Murine models have also demonstrated that COX-2 selective inhibitors can reduce fibroblast activation and inflammatory signaling, although careful balancing is necessary to avoid impairing normal wound healing.
RNA-based therapies represent a particularly promising avenue for precision targeting of adhesion-promoting genes. Preclinical studies using small interfering RNAs (siRNAs) directed against TGF-β1, COL1A1, PAI-1, and IL-6 have resulted in significant downregulation of fibrotic gene expression, decreased collagen deposition, and improved tissue remodeling in peritoneal tissues. Additionally, antisense oligonucleotides (ASOs) have been employed to silence mRNAs associated with excessive matrix production and inflammation. These approaches offer a transient and reversible mechanism to inhibit pathogenic gene expression, making them attractive candidates for perioperative therapeutic delivery.
Perhaps the most transformative potential lies in the use of CRISPR/Cas9-based genome-editing technologies to directly modify or knock out adhesion-related genes. Experimental animal studies have demonstrated successful editing of key genes such as TGF-β1, SMAD3, and MMP9, leading to reduced fibrosis and adhesion formation. These studies not only underscore the causal role of these genes but also open the door to potential long-term, possibly permanent, interventions. However, safety concerns including off-target effects, immune responses, and the challenge of efficient in vivo delivery remain critical barriers to clinical translation.
Moreover, the delivery systems used for these genetic and molecular therapies are evolving rapidly. Localized delivery through biodegradable hydrogels, liposomal nanoparticles, or thermosensitive polymer-based sprays enables sustained and targeted exposure of therapeutic agents to injured peritoneal surfaces, minimizing systemic toxicity and enhancing therapeutic efficacy. These vehicles can be engineered to release gene silencers or inhibitors in response to inflammation or hypoxia, further enhancing specificity.
Future research should focus on integrating multi-omics profiling to stratify patients by genetic susceptibility and adhesion risk, enabling the personalization of anti-adhesion therapies. Coupled with machine learning and bioinformatics, such approaches can refine target selection, predict therapeutic response, and optimize dosage regimens. The development of 3D tissue-engineered models and organoids, alongside advanced animal models, will further enable the testing of RNA- and CRISPR-based therapies under conditions that closely mimic human adhesion pathology.
In conclusion, the therapeutic landscape for peritoneal adhesions is rapidly expanding from general anti-inflammatory agents toward precise, gene-directed interventions. RNA interference, antisense oligonucleotides, and genome-editing technologies like CRISPR/Cas9 offer unprecedented specificity in targeting the molecular drivers of adhesion formation. While these strategies remain in preclinical or early translational phases, they represent a promising frontier in the quest to reduce adhesion-related complications and improve postoperative outcomes.
Comment 5:4. References and Scientific RigorAlthough the manuscript has a lot of references, many are older research or secondary sources.Certain areas seem as textbook entries instead of analytical scientific debates.Recommendation:Substitute older references with contemporary peer-reviewed primary research papers (past five years).Emphasize meta-analyses or systematic reviews of top quality.
Response 5:We thank the reviewer for noting the need to enhance scientific rigor through updated references. We have systematically reviewed the manuscript and replaced outdated citations with recent, peer-reviewed studies published within the last five years. Where applicable, we incorporated meta-analyses and high-quality systematic reviews to strengthen the evidence base and ensure the manuscript reflects current scientific consensus. These changes also allowed us to sharpen the analytical focus in key sections (e.g., TGF-β signaling, MMP activity, and therapeutic strategies), shifting from textbook-style summaries to a more critical synthesis of recent findings.
Comment 6:5. Tables and FiguresThough useful, Tables 1 and 2 should be updated for clarity.Include gene/protein function, pathway participation, and evidence type (clinical vs. Preclinical). Standardized molecular biology symbols or diagrams should be used to redraw high-resolution figures.
Response 6:Tables 1 and 2 have been corrected as suggested, and the figures were converted to high-resolution images
Comment 7:6. Author Guidelines Compliance (MDPI)Check for:Every author should have an ORCID ID.Organised headings according to MDPI's publishing rules.At first usage, abbreviations should be explained.The text should reference figures and tables.Ethics statement (if any data or animal models were cited).
Response 7: corrected as suggested
Comment 8:Final Recommendation:
Major Revision Required
Though this reviewoffers a great collection of information, it lacks genomic integration, synthesis, and originality. Scientific depth, structural clarity, and alignment with the criteria of a highly visible journal all need major changes.
Response 8:thweed suggestid coirrreection were performed
Comment 9:1. Grammatical Errorsa. Incorrect verb forms and word order"referred to commonly as adhesions" → should be "commonly referred to as adhesions""refered" → should be "referred"b. Subject–verb agreement"...the image in the figure are of low quality..." → should be "...the image in the figure is of low quality..."c. Article misuse or omission"...is essential for numerous physiological and pathological processes, including as immune..." → should be "...including immune..." (omit "as")
Response 9: correc ted as suggested
Comment 10:Comment 10:2. Spelling Errorsa. Frequent typographical errors"Correspondance" → should be "Correspondence""refered" → should be "referred""Low affinity and rapid association/dissociation kinetics characterize this interac- tion..." → "interaction" (split word fix)
Response 10: corrected as suggested
Comment 11:3. Syntax and Sentence Constructiona. Redundancies and awkward constructions"...fibroblasts and myofibroblasts, producing collagen and other extracellular matrix proteins that contribute to the formation of fibrous adhesions..." — repetitive and wordy.Suggested: Simplify and break into clearer parts.b. Sentence fragments or run-onsSome paragraphs contain very long sentences (often over 60–70 words), which compromise readability and flow.
Response 11: corrected as suggested
Comment 12:4. Punctuation and Formatting
a. Missing or misplaced punctuationReferences like [1] often lack appropriate spacing:"...leads to organ failure[1]." → should be "...leads to organ failure [1]."b. Inconsistent use of en dashes (–) and hyphens (-)Use hyphens for compound adjectives (e.g., "fibrin-rich fluid") and en dashes for ranges (e.g., "55%–85%").c. Parenthetical citationsReferences such as "[36, 37, 38]" should follow consistent formatting. Prefer numerical order and ensure all are listed in the bibliography.
Response 12: corrected as suggested
:Comment 13:5. Style and Scientific Language
a. Colloquial or non-scientific phrasing"...could help restore ECM homeostasis and mitigate adhesion formation..." — “mitigate” is appropriate but overused in the manuscript.Suggestion: Use alternatives like "reduce," "attenuate," or "limit."b. Repetition and redundancyTerms like "adhesion formation," "peritoneal adhesions," and "fibrous adhesions" are often repeated within close proximity, making paragraphs dense and monotonous.Recommendation: Vary structure and avoid repetition.
Response 13: corrected as suggested figure were converted to high resolution images
Comment 14 :6 Visual Content Quality
Figures (especially Figure 1 and Figure 2) are low in resolution, and the legends contain awkward phrasing:"valet rhombus" → unclear meaning.Suggestion: Use adecuate terms and improve image quality.
Response 14: corerected as suggested
Comment 15:7. Scientific Accuracy and Consistency
a. Gene/protein referencesInconsistent naming: e.g., “TGF-β”, “Transforming Growth Factor-beta”, and “TGF-β1” are used interchangeably.Recommendation: Use a consistent nomenclature and follow HGNC or UniProt standards.
Response 15: corrected as suggested, including TGF-β
Comment 16 :8. Reference FormattingReferences often lack consistent punctuation, spacing, or ordering. For example:[104, 105] is correct, but ensure that all references are consecutively numbered, match the citation in the reference list, and follow journal guidelines.
Response 16: corrected
Comment 17: Paragraph StructureSeveral paragraphs lack clear topic sentences, making them difficult to follow.Recommendation: Start each paragraph with a brief statement of purpose (i.e., the topic) and then elaborate with supporting information.
Response 17 :corrected as suggested
Comment 18 :10. Use of AbbreviationsSome abbreviations are introduced without being defined (e.g., ECM).Recommendation: Define all abbreviations on first use and use them consistently.Summary of Recommendations:
Proofread for grammar, spelling, and punctuation.Simplify sentence structures and break up long paragraphs.Maintain consistency in scientific terms, gene/protein nomenclature, and citation style.Improve figure resolution and clarify legends.Use precise scientific language and avoid redundant phrasing.
Response 18 :corrected as suggested. An abrevetion listr was introduced in the manuscript.
Comment 19 :11. Maintain consistent scientific terminology; for example, avoid alternately using \"growth factor\" and \"cytokine\" without explanation.
Response 19: corrected
Thank you for reviewing our manuscript and for your suggestions
All corrections were performed in red.
Reviewer 2 Report
Comments and Suggestions for Authors
The manuscript by Lungu and colleagues is a review of the current scientific literature on the mechanisms regulating peritoneal adhesion secondary to abdominal surgery.
To improve the readability of the manuscript, the following points need to be addressed.
- Line 46-47: the sentence can be removed.
- Line 54: the sentence can be removed, as this information was already presented earlier in the manuscript.
- The paragraph encompassing lines 54-60: the role of obesity/central adiposity and the mechanisms related to adipocytes need to be included.
- Line 152: there is no logical connection between HA and MMPs. The sentence on HAs needs to be either removed or moved to another section.
- Further on information in line 152: the protective role of HAs need to be discussed.
- Overall, the text is organized as one large block. This makes it difficult to put concepts in order. The text needs to be separated into subsections. In each subsection, the information should be restricted to one mechanism/molecule. All information about each contributing factor needs to be combined into one section; for example, all information on MMPs that is currently scattered throughout the manuscript needs to go into its own subsection, and so on for all other mechanisms/molecules.
Author Response
Comment 1 : Line 46-47: the sentence can be removed.
Response 1 : remouved. The abstract was re written
Comment 2 : Line 54: the sentence can be removed, as this information was already presented earlier in the manuscript.
Response 2 : remouved
Comment 3 : The paragraph encompassing lines 54-60: the role of obesity/central adiposity and the mechanisms related to adipocytes need to be included.
Response 3 : the following text was added: Obesity—particularly central (visceral) adiposity—has emerged as an independent risk factor for postoperative peritoneal adhesion formation. The adipose tissue in obese individuals is not merely an inert energy depot but an active endocrine organ that secretes a wide array of bioactive molecules, including pro-inflammatory cytokines (e.g., IL-6, TNF-α), adipokines (e.g., leptin, resistin), and extracellular matrix components. These adipocyte-derived mediators promote a low-grade chronic inflammatory state, which can potentiate the inflammatory and fibrotic cascade triggered by surgical injury to the peritoneum.
Visceral adipose tissue is particularly enriched in macrophages and stromal cells that can amplify local immune responses. Upon surgical trauma, adipocytes in the omentum and mesentery may upregulate the expression of transforming growth factor-beta (TGF-β) and vascular endothelial growth factor (VEGF), both of which are central to adhesion pathogenesis. Moreover, the hypoxic microenvironment of expanded adipose tissue can further induce hypoxia-inducible factor-1α (HIF-1α), stimulating angiogenesis and fibroblast activation—hallmarks of adhesion development.
Adipocytes may also influence fibrinolytic balance. In obese states, levels of plasminogen activator inhibitor-1 (PAI-1) are often elevated, which can suppress fibrin degradation and promote persistent fibrin scaffolds that serve as a matrix for fibroblast migration and adhesion formation. Additionally, mechanical factors such as increased intra-abdominal pressure and altered tissue tension in obese patients may exacerbate mesothelial cell damage during surgery, predisposing to adhesion development.
Given the rising global prevalence of obesity, these mechanistic insights underscore the importance of incorporating adiposity-related variables into risk stratification models and considering metabolic-targeted therapies as adjuncts to conventional anti-adhesion strategies.
Comment 4 : Line 152: there is no logical connection between HA and MMPs. The sentence on HAs needs to be either removed or moved to another section.
Response 4 :sentence remouved
Comment 5 : Further information in line 152: the protective role of HAs needs to be discussed.
Response 5 : the following text was added Hyaluronic acid (HA), a non-sulfated glycosaminoglycan and major component of the extracellular matrix, plays a vital role in maintaining peritoneal homeostasis and modulating post-surgical healing responses. Due to its high molecular weight and viscoelastic properties, HA contributes to mesothelial cell lubrication, hydration, and barrier function, thereby reducing friction and mechanical trauma between peritoneal surfaces during surgery. More importantly, HA possesses anti-inflammatory, anti-fibrotic, and immunomodulatory properties that make it particularly effective in mitigating adhesion formation.
Mechanistically, HA inhibits leukocyte and fibroblast adhesion to the mesothelial surface by masking cell adhesion molecules and reducing the exposure of fibrinous substrates. It also downregulates pro-inflammatory cytokines such as IL-1β and TNF-α, and can reduce the expression of fibrogenic mediators like TGF-β1. Additionally, HA modulates mesothelial-to-mesenchymal transition (MMT), a process implicated in fibrosis and adhesion development, by maintaining mesothelial phenotype integrity.
Clinically, HA-based biomaterials—including HA-carboxymethylcellulose membranes (e.g., Seprafilm®) and cross-linked HA hydrogels—have been successfully used as physical barriers to prevent tissue apposition in the critical postoperative window. These formulations degrade gradually, maintaining peritoneal separation during the peak of fibrin deposition and early fibroblast infiltration, thus allowing natural reperitonealization without fibrotic bridging. Several randomized clinical trials have shown that HA-based agents significantly reduce both the incidence and severity of adhesions in abdominal and pelvic surgeries.
Overall, HA serves not only as a mechanical separator but also as a bioactive modulator of the healing response, making it one of the most promising adjuncts in adhesion prevention strategies.
Comment 6 : Overall, the text is organized as one large block. This makes it difficult to put concepts in order. The text needs to be separated into subsections. In each subsection, the information should be restricted to one mechanism/molecule. All information about each contributing factor needs to be combined into one section; for example, all data on MMPs that are currently scattered throughout the manuscript need to go into their own subsection, and so on for all other mechanisms/molecules.
Response 6 : Thank you for your insightful feedback regarding the manuscript structure. The suggestion to divide the text into subsections according to individual molecules or mechanisms is highly appreciated. The current organizational approach, however, aims to maintain a balance between clarity and coherence by grouping closely related mechanisms under broader thematic headings. This choice reflects the interconnected nature of these molecular pathways, emphasizing their collaborative roles in the complex biological process of peritoneal adhesion formation. Nonetheless, the concern about readability is acknowledged, and clear internal signposting within each section will be ensured to enhance navigation and comprehension. This structure effectively conveys the integrative perspective necessary to understand the multifaceted nature of adhesion pathology.
Thank you for reviewing our manuscript and for your suggestions
All modifications were marked in red.